# Multimorbidity and adverse events of special interest associated with Covid-19 vaccines in Hong Kong

Francisco Tsz Tsun Lai [1,2,10], Lei Huang[1,10], Celine Sze Ling Chui[2,3,4], Eric Yuk Fai Wan[1,2,5], Xue Li[2,6], Carlos King Ho Wong [1,2,5], Edward Wai Wa Chan[1], Tiantian Ma[1,2], Dawn Hei Lum[1], Janice Ching Nam Leung[1,2], Hao Luo[7,8], Esther Wai Yin Chan[1,2] & Ian Chi Kei Wong [1,2,9✉]

Prior research using electronic health records for Covid-19 vaccine safety monitoring typically focuses on specific disease groups and excludes individuals with multimorbidity, defined as ≥2 chronic conditions. We examine the potential additional risk of adverse events 28 days after the first dose of CoronaVac or Comirnaty imposed by multimorbidity. Using a territory-wide public healthcare database with population-based vaccination records in Hong Kong, we analyze a retrospective cohort of patients with chronic conditions. Thirty adverse events of special interest according to the World Health Organization are examined. In total, 883,416 patients are included and 2,807 (0.3%) develop adverse events. Results suggest vaccinated patients have lower risks of adverse events than unvaccinated individuals, multimorbidity is associated with increased risks regardless of vaccination, and the association of vaccination with adverse events is not modified by multimorbidity. To conclude, we find no evidence that multimorbidity imposes extra risks of adverse events following Covid-19 vaccination.

[1] Centre for Safe Medication Practice and Research, Department of Pharmacology and Pharmacy, Li Ka Shing Faculty of Medicine, The University of Hong Kong, Hong Kong Special Administrative Region, China. [2] Laboratory of Data Discovery for Health (D24H), Hong Kong Science Park, Hong Kong Science and Technology Park, Hong Kong Special Administrative Region, China. [3] School of Nursing, Li Ka Shing Faculty of Medicine, The University of Hong Kong, Hong Kong Special Administrative Region, China. [4] School of Public Health, Li Ka Shing Faculty of Medicine, The University of Hong Kong, Hong Kong Special Administrative Region, China. [5] Department of Family Medicine and Primary Care, Li Ka Shing Faculty of Medicine, The University of Hong Kong, Hong Kong Special Administrative Region, China. [6] Department of Medicine, Li Ka Shing Faculty of Medicine, The University of Hong Kong, Hong Kong Special Administrative Region, China. [7] Department of Social Work and Social Administration, Faculty of Social Sciences, The University of Hong Kong, Hong Kong Special Administrative Region, China. [8] Department of Computer Science, Faculty of Engineering, The University of Hong Kong, Hong Kong Special Administrative Region, China. [9] Research Department of Practice and Policy, School of Pharmacy, University College London, London, UK. [10] These authors contributed equally: Francisco Tsz Tsun Lai, Lei Huang. ✉email: wongick@hku.hk

The safety of Covid-19 vaccines is of great public health concern and is crucial to tackling vaccine hesitancy amidst the pandemic[1]. In particular, there have been widespread speculations of cardiovascular and other adverse events of special interest (AESI) in relation to Covid-19 vaccines[2,3]. This may be due to thromboembolic safety signals[4,5] and case reports of other adverse outcomes, such as Bell's palsy[6,7] and myocarditis[8,9] following the administration of specific vaccines.

There is also increased safety concern regarding the vaccination of people living with underlying chronic conditions[10–13] and multimorbidity[14], commonly referred to as the co-occurrence of two or more chronic health conditions in an individual[15]. Previous research before the pandemic has shown a potential risk increase of cardiovascular events and other adverse outcomes in people living with multimorbidity compared with those without[16,17]. Specifically, the complex underlying mechanisms of the co-occurrence of multiple health conditions and multiple medications have been found to be related to increased adverse drug reactions[18–20]. Established evidence shows that varying degrees of inflammation may typically be induced by vaccination and the associated immune responses in general[21], cautious pharmacovigilance is indeed warranted for people with multimorbidity who are at a higher risk of adverse health outcomes to begin with. Of note, people living with multimorbidity are at increased risk of serious complications following an infection of SARS-CoV-2[22]. It is currently unclear if multimorbidity is related to a risk increase of any AESI following Covid-19 vaccination. Nevertheless, existing research comparing the relationship between vaccination and AESI across sub-populations with and without multimorbidity is limited, rendering the risk and benefit assessment for the vaccination of the multimorbid populations inconclusive.

Hong Kong is one of the relatively few jurisdictions in the world that has approved and rolled out the widespread emergency use of both CoronaVac (Sinovac)[23] and Comirnaty (Fosun-BioNTech, equivalent to Pfizer-BioNTech outside China)[24] Covid-19 vaccines[25]. We analyzed the territory-wide public healthcare databases linked with population-based vaccination records from the Government to examine the risk of AESI of these two vaccines. This study aims to examine the relationship between the first dose of Covid-19 vaccination and AESI among patients with chronic disease in Hong Kong and the potential additional AESI risk following vaccination associated with multimorbidity.

## Results

As shown in Fig. 1, among 3,983,529 patients who used the Hospital Authority (HA) services, 1,643,419 (41.3%) were vaccinated (at least one dose). In all, 1,391,033 patients were identified as having at least one diagnosis of any of the 20 listed chronic conditions. After age- and sex-matching for the mapping of the index date from the vaccinated to the unvaccinated group, 1,184,476 patients remained. Eventually, 883,416 patients were adopted as the final cohort with 38.0% of the patients vaccinated, after a further removal of ineligible patients. The mean follow-up time [standard deviation (SD)] for the CoronaVac ($n = 182,442$), Comirnaty ($n = 153,178$), and unvaccinated groups ($n = 547,796$) were 23.60 (8.17), 19.14 (6.57), and 23.14 (8.45) days respectively.

**Cohort characteristics**. Table 1 shows the cohort characteristics before and after weighting. The unvaccinated group had the highest mean age of 62.11 (SD: 12.85) followed by 61.58 (SD: 11.08) among the CoronaVac group and 56.81 (SD:13.43) among the Comirnaty group. There were higher proportions of men in the vaccinated groups (48.7% for CoronaVac; 47.4% for

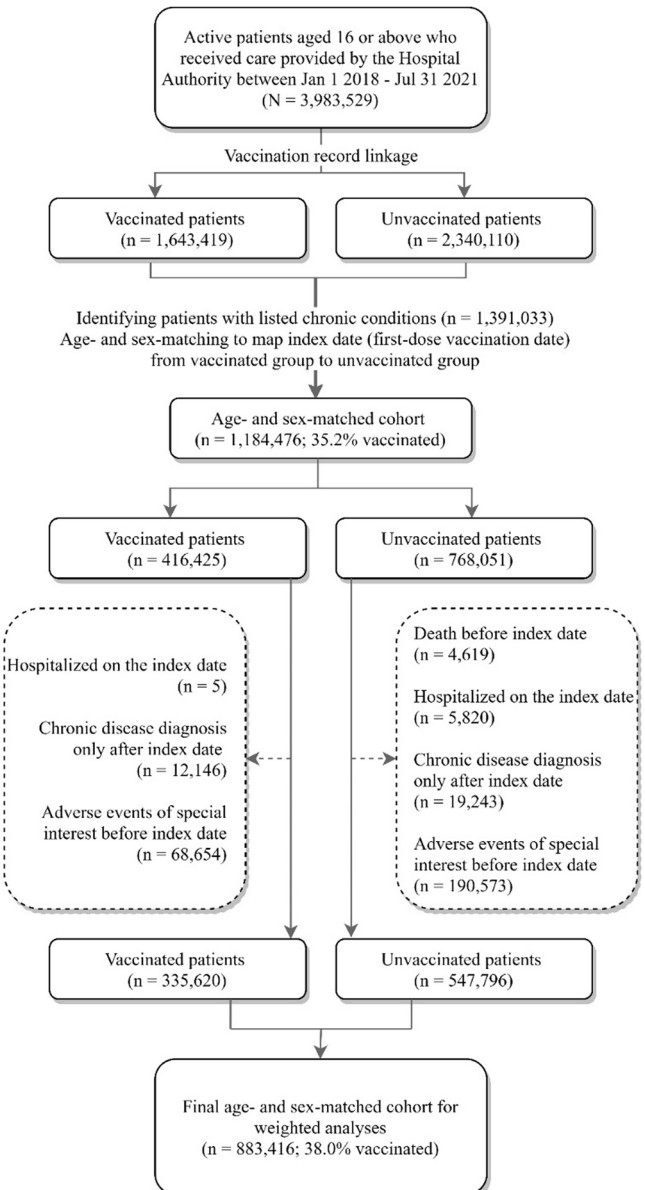

**Fig. 1 Flow chart of cohort selection.** A total of 883,416 patients were included in the final cohort with 38.0% of the patients vaccinated.

Comirnaty) than in the unvaccinated group (41.9%). For all three groups, the most prevalent condition was hypertension (67.1% for unvaccinated; 68.7% for CoronaVac; 60.9% for Comirnaty), followed by diabetes (type 2) (33.0% for unvaccinated group; 28.1% for CoronaVac; 23.8% for Comirnaty), and severe constipation (8.3% for unvaccinated group; 9.0% for CoronaVac; 9.0% for Comirnaty). The entropy rebalancing technique we adopted produced a series of appropriate weights to achieve the highest possible balance between the three groups without using anyone of the groups as referent. After weighting, the maximum standardized mean differences (SMD) for all covariates were all <0.1. Supplementary Table 1 shows the frequencies of patients by age and multimorbidity status. Unsurprisingly, multimorbidity was shown to be more prevalent among older age groups.

**Adverse events of special interest**. During the observation period, a total of 2807 (0.3%) patients had AESI: 2046 were among the unvaccinated group (0.4%), 469 were among the CoronaVac

**Table 1 Unweighted and weighted cohort characteristics (N = 883,416) of unvaccinated individuals, CoronaVac, or Comirnaty recipients.**

| | Unvaccinated Unweighted | CoronaVac | Comirnaty | Unvaccinated Weighted[a] | CoronaVac | Comirnaty | |
|---|---|---|---|---|---|---|---|
| n | 547,796 | 182,442 | 153,178 | 536,072 | 124,524 | 96,322 | Maximum weighted SMD |
| Age [mean (SD)] | 62.11 (12.85) | 61.58 (11.08) | 56.81 (13.43) | 61.08 (13.33) | 61.08 (11.31) | 61.08 (12.17) | <0.001 |
| Sex: male (%) | 229,791 (41.9) | 88,881 (48.7) | 72,586 (47.4) | 242587.9 (44.3) | 80814.1 (44.3) | 67831.5 (44.3) | <0.001 |
| Chronic conditions (%) | | | | | | | |
| Hypertension | 367,799 (67.1) | 125,314 (68.7) | 93,319 (60.9) | 363621.4 (66.4) | 121104.0 (66.4) | 101688.5 (66.4) | <0.001 |
| Diabetes (type 2) | 180,875 (33.0) | 51,312 (28.1) | 36,403 (23.8) | 166524.0 (30.4) | 55471.7 (30.4) | 46570.1 (30.4) | <0.001 |
| Severe constipation | 45,405 (8.3) | 16,424 (9.0) | 13,804 (9.0) | 46915.4 (8.6) | 15619.6 (8.6) | 13111.6 (8.6) | <0.001 |
| Depression | 34,333 (6.3) | 10,745 (5.9) | 13,310 (8.7) | 36257.6 (6.6) | 12056.4 (6.6) | 10126.4 (6.6) | <0.001 |
| Cancer | 31,308 (5.7) | 4671 (2.6) | 4733 (3.1) | 25244.0 (4.6) | 8407.6 (4.6) | 7050.6 (4.6) | <0.001 |
| Hypothyroidism | 27,752 (5.1) | 9555 (5.2) | 9508 (6.2) | 29019.7 (5.3) | 9673.0 (5.3) | 8119.2 (5.3) | <0.001 |
| Chronic pain | 23,305 (4.3) | 7577 (4.2) | 7845 (5.1) | 24013.3 (4.4) | 8006.3 (4.4) | 6699.9 (4.4) | <0.001 |
| Asthma | 18,172 (3.3) | 5119 (2.8) | 7061 (4.6) | 18773.8 (3.4) | 6277.3 (3.4) | 5265.2 (3.4) | <0.001 |
| Chronic pulmonary disease | 12,090 (2.2) | 2638 (1.4) | 1663 (1.1) | 10152.9 (1.9) | 3373.7 (1.8) | 2846.7 (1.9) | <0.001 |
| Schizophrenia | 10,915 (2.0) | 1634 (0.9) | 1637 (1.1) | 8803.3 (1.6) | 2930.7 (1.6) | 2463.7 (1.6) | <0.001 |
| Rheumatoid arthritis | 7654 (1.4) | 1582 (0.9) | 1663 (1.1) | 6783.9 (1.2) | 2257.3 (1.2) | 1883.2 (1.2) | <0.001 |
| Peptic ulcer disease | 6662 (1.2) | 2224 (1.2) | 1583 (1.0) | 6502.3 (1.2) | 2162.0 (1.2) | 1810.1 (1.2) | <0.001 |
| Alcohol misuse | 4535 (0.8) | 1356 (0.7) | 1278 (0.8) | 4441.6 (0.8) | 1468.2 (0.8) | 1242.3 (0.8) | <0.001 |
| Cirrhosis | 3294 (0.6) | 544 (0.3) | 433 (0.3) | 2634.9 (0.5) | 876.7 (0.5) | 746.0 (0.5) | <0.001 |
| Parkinson's disease | 3182 (0.6) | 409 (0.2) | 336 (0.2) | 2419.1 (0.4) | 811.4 (0.4) | 685.1 (0.4) | <0.001 |
| Dementia | 3030 (0.6) | 299 (0.2) | 204 (0.1) | 2184.8 (0.4) | 734.9 (0.4) | 607.3 (0.4) | 0.001 |
| Psoriasis | 2524 (0.5) | 721 (0.4) | 814 (0.5) | 2513.0 (0.5) | 836.6 (0.5) | 705.5 (0.5) | <0.001 |
| Irritable bowel syndrome | 1710 (0.3) | 621 (0.3) | 823 (0.5) | 1960.3 (0.4) | 652.3 (0.4) | 545.5 (0.4) | <0.001 |
| Inflammatory bowel disease | 1548 (0.3) | 360 (0.2) | 595 (0.4) | 1584.6 (0.3) | 514.3 (0.3) | 427.3 (0.3) | 0.002 |
| Peripheral vascular disease | 93 (0.0) | 9 (0.0) | 9 (0.0) | 91.5 (0.0) | 8.7 (0.0) | 13.9 (0.0) | 0.012 |
| Multimorbidity status[b] (%) | | | | | | | 0.012 |
| One condition | 344,866 (63.0) | 128,890 (70.6) | 114,311 (74.6) | 363464.1 (66.4) | 121971.7 (66.9) | 102332.2 (66.8) | – |
| Two conditions | 168,166 (30.7) | 46,844 (25.7) | 34,071 (22.2) | 155986.2 (28.5) | 51013.4 (28.0) | 42906.3 (28.0) | – |
| Three conditions | 28,767 (5.3) | 5893 (3.2) | 4224 (2.8) | 23974.7 (4.4) | 7938.7 (4.4) | 6712.3 (4.4) | – |
| Four or more conditions | 5997 (1.1) | 815 (0.4) | 572 (0.4) | 4371.0 (0.8) | 1518.2 (0.8) | 1227.3 (0.8) | – |
| Adverse events of special interest[b] (%) | | | | | | | |
| Cardiovascular system | 759 (0.1) | 160 (0.1) | 106 (0.1) | 737.8 (0.1) | 158.1 (0.1) | 118.0 (0.1) | – |
| Circulatory system | 689 (0.1) | 135 (0.1) | 113 (0.1) | 664.4 (0.1) | 131.6 (0.1) | 134.3 (0.1) | – |
| Hepato-renal system | 513 (0.1) | 148 (0.1) | 86 (0.1) | 508.3 (0.1) | 147.9 (0.1) | 85.7 (0.1) | – |
| Auto-immune diseases | 229 (0.0) | 40 (0.0) | 55 (0.0) | 233.5 (0.0) | 41.3 (0.0) | 50.6 (0.0) | – |
| Respiratory system | 154 (0.0) | 18 (0.0) | 10 (0.0) | 140.9 (0.0) | 17.9 (0.0) | 12.9 (0.0) | – |
| Other system | 113 (0.0) | 42 (0.0) | 43 (0.0) | 105.9 (0.0) | 43.3 (0.0) | 33.7 (0.0) | – |
| Nerves and central nervous system | 63 (0.0) | 14 (0.0) | 16 (0.0) | 59.5 (0.0) | 13.6 (0.0) | 16.6 (0.0) | – |
| Skin, bone, and joints system | 2 (0.0) | 1 (0.0) | 4 (0.0) | 1.7 (0.0) | 1.1 (0.0) | 2.9 (0.0) | – |

SMD standardized mean difference.
[a] Weighting variables: age, sex, hypertension, diabetes mellitus (type 2), severe constipation, depression, cancer, hypothyroidism, chronic pain, asthma, alcohol misuse, chronic pulmonary disease, schizophrenia, rheumatoid arthritis, peptic ulcer disease, cirrhosis, psoriasis, Parkinson's disease, dementia, irritable bowel syndrome, inflammatory bowel disease, and peripheral vascular disease.
[b] Not included for weighting.

group (0.3%), and 292 were among the Comirnaty group (0.2%). The incidence rates for the unvaccinated, CoronaVac, and Comirnaty groups were 59.0 (95% CI 56.4–61.5), 39.8 (95% CI 36.2–43.4), and 36.4 (95% CI 32.2–40.6) per 1000 person-years respectively. Figure 2 shows the cumulative incidence of AESI by multimorbidity status and vaccine group over the follow-up period. Patients with multimorbidity were observed to have a faster increase of AESI incidence but no marked differences were identified between vaccine groups. Figure 3 shows three-chord diagrams by vaccine group exemplifying the relative frequencies (represented by ribbon area) of AESI-chronic condition pairings with each color representing a specific AESI. The pairings were similarly patterned across all three groups, suggesting the co-

occurrence of specific chronic conditions and AESI are similar between the unvaccinated, those receiving Comirnaty and those receiving CoronaVac. The underlying frequencies of the diagrams are shown in Supplementary Table 2. The frequencies of specific AESI by vaccine group were tabulated as Supplementary Table 3.

**Cox proportional hazard model.** As shown in Table 2, Model 1 of the Cox proportional hazard regression analysis suggested that patients who received vaccines had a lower risk of AESI [hazard ratio (HR) = 0.66, 95% CI 0.58–0.75 for Comirnaty and HR = 0.70, 95% CI 0.63–0.77 for CoronaVac]. Model 2 suggested that multimorbidity was associated with 63%-increased hazards of

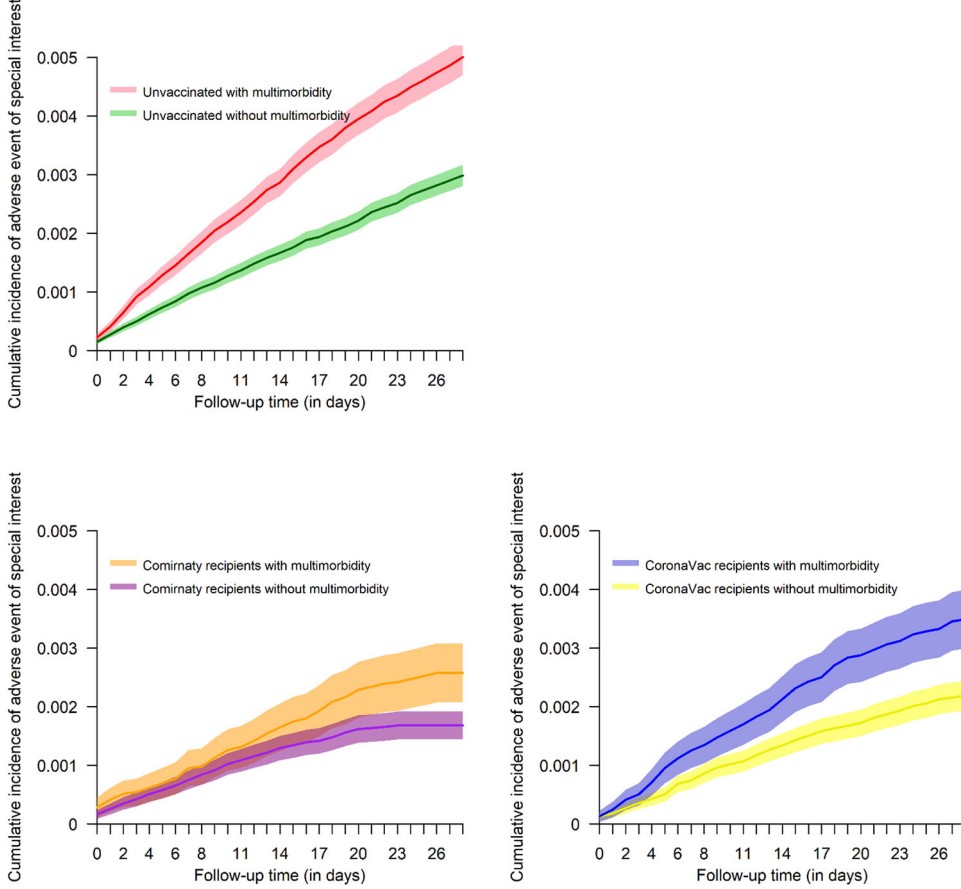

**Fig. 2 Cumulative incidence of adverse events of special interest over the follow-up period represented by the lines with 95% confidence intervals represented by the shaded area.** Different colors represent groups of different multimorbidity status and vaccine group.

AESI (HR = 1.63, 95% CI 1.51–1.75). Model 3 suggested there was no significant modification of AESI risk associated with vaccine group by multimorbidity (HR = 0.88, 95% CI 0.67–1.15 for Comirnaty; HR = 1.03, 95% CI 0.84–1.27 for CoronaVac). For analyses on sub-categories of AESI, results were largely similar with the main findings with differing degrees of variation of the point estimates (Supplementary Table 4). Supplementary Table 5 shows the results from Model 3 for all specific AESIs, which were in line with the main findings. We did not estimate the HR for AESIs of which only five or fewer cases occurred in any of the vaccine groups to avoid misinterpretation.

**Sensitivity analyses.** No marked deviations from the main analysis were observed in the findings of a series of sensitivity analysis although there were variations in the point estimates of the associations (Supplementary Tables 6–14). However, conclusions from the findings were not affected.

## Discussion

We found no evidence of a modified association between vaccination (first dose) and AESI among those living with multimorbidity compared with those without in general. This finding was also true for sub-categories of AESI. Regardless of vaccination, there is a significantly heightened risk among people with multimorbidity compared with those having only one condition.

Our results showed a lower risk of AESI among patients receiving the vaccines than among those who did not, even after we excluded patients with a hospitalization record in the past

6 months from the analyses. This finding may reflect a healthy user bias whereby patients who decided to get vaccinated were those who had their chronic conditions better controlled even given the same diagnoses[26]. This observation is in line with the official guidelines published by the Hong Kong Government only recommending that patients living with chronic conditions receive the vaccine if their conditions are under stable control[27]. Our results may not be applicable to recipients who had poorly controlled chronic conditions. In fact, the vaccination rate among the age group of ≥80 was as low as 16.9% as of 9 November 2021[28]. Nevertheless, even with this potential bias towards an inverse association of vaccines with AESI, there should be no impact on our key result of no stronger association of vaccines with AESI across multimorbidity status. This is because this bias should apply to both patients with multimorbidity and patients with only one listed condition. If multimorbidity does impose additional AESI risk following vaccination, the test for effect modification (interaction in Model 3) should still be able to detect this risk increase.

These findings largely agree with the existing published data on the overall safety profile of the two investigated Covid-19 vaccines, suggesting no significant safety signals of an increased risk of AESI overall[6,29], except recent studies on a heightened risk of very rare diseases, including Bell's palsy following the use of CoronaVac[7], myocarditis[8], and hemorrhagic stroke[30] following Comirnaty. Nonetheless, current post-marketing research in this regard is still limited and accruing[29]. In fact, to the best of our knowledge, no research has examined the role of multimorbidity in the potential risk elevation of AESI. In a protocol template for

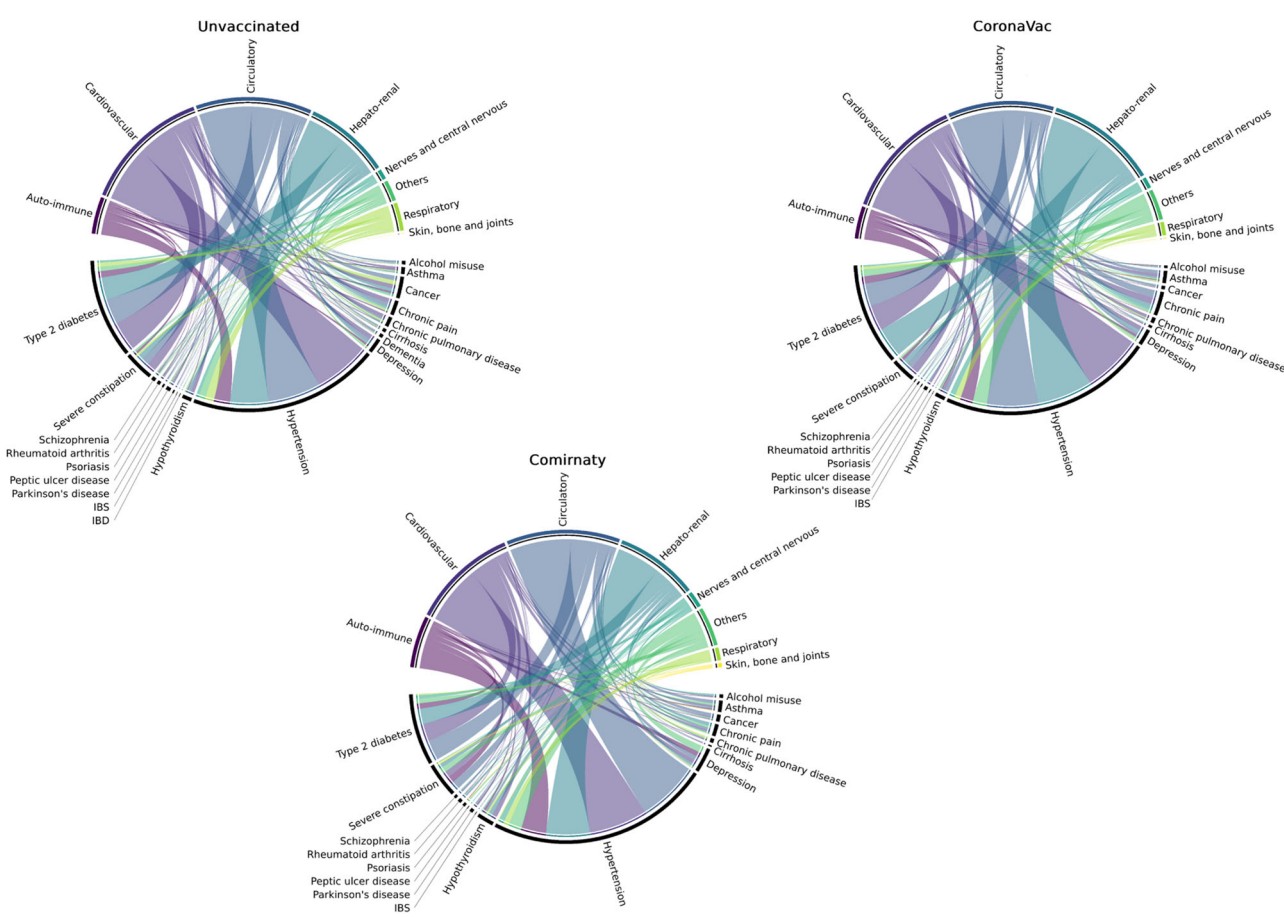

**Fig. 3 Chord diagrams showing the relative frequencies of the co-occurrence of specific chronic disease (lower arc) and specific adverse events of special interest (upper arc) by vaccine group.** The larger the area of the chord linking between a chronic condition and an adverse event of special interest, the more frequently observed the co-occurrence of them. IBS irritable bowel syndrome, IBD inflammatory bowel disease.

**Table 2 Hazard ratios with 95% confidence intervals (CI) of adverse events of special interest generated from Cox proportional hazard models with entropy rebalancing.**

|  | Model 1 | Model 2 | Model 3 |
|---|---|---|---|
|  | Hazard ratios (95% CI), *P*-value |  |  |
| Vaccine group |  |  |  |
| Unvaccinated | Ref. | Ref. | Ref. |
| Comirnaty | 0.66 (0.58, 0.75), < 0.001 | 0.66 (0.58, 0.75), < 0.001 | 0.70 (0.59, 0.82), < 0.001 |
| CoronaVac | 0.70 (0.63, 0.77), < 0.001 | 0.70 (0.63, 0.78), < 0.001 | 0.69 (0.61, 0.79), < 0.001 |
| Multimorbidity status |  |  |  |
| One chronic condition | – | Ref. | Ref. |
| Multimorbid | – | 1.63 (1.51, 1.75), < 0.001 | 1.64 (1.50, 1.79), < 0.001 |
| Interaction |  |  |  |
| Comirnaty X multimorbidity | – | – | 0.88 (0.67, 1.15), 0.332 |
| CoronaVac X multimorbidity | – | – | 1.03 (0.84, 1.27), 0.792 |

Included independent variables in Model 1: vaccine group only; Model 2: Model 1 + multimorbidity status; Model 3: Model 2 + interaction between vaccine group and multimorbidity.

electronic healthcare databases monitoring under the vACCine covid-19 monitoring readinESS (ACCESS) project funded by the European Medicines Agency[31], it was recommended that specific at-risk disease groups be examined individually with multimorbidity excluded from analyses. While this recommended approach may contribute to more specific information about the risk profile of vaccines for specific disease groups, it is far from ideal to disregard the presence of a significant proportion of patients with more than one condition. According to a systematic

review, global community prevalence of multimorbidity is estimated at approximately one-third[32]. Any research excluding multimorbid patients has limited generalizability to this significant proportion of populations. As far as we are aware, this is the first post-marketing pharmacovigilance study testing for a potential AESI risk elevation associated with multimorbidity. Interestingly, we observed when the analysis was confined to AESI recorded in the inpatient setting, the increase of AESI risks associated with multimorbidity regardless of vaccination was

greater although the confidence intervals overlapped and differences were non-significant. This may be because those AESI requiring tertiary care were more severe than the more broadly defined ones.

Subject to further international research to replicate and verify our results, the implications of this study are important to reassure the public with regard to the widespread concern about vaccine safety among individuals living with multimorbidity who might be hesitant towards vaccine uptake[33]. First, the incidence of AESI was rare even among a cohort of 883,416, with an incidence rate of 51.5 (95% CI 49.6–53.4) per 1000 person-years. Second, we showed that although multimorbidity was associated with a higher risk of AESI, this association was independent of Covid-19 vaccination. Given the fact that people with multimorbidity have a higher risk of developing life-threatening complications if infected with SARS-CoV-2[34], our results should be reassuring and support the notion that multimorbidity does not impose an additional risk of AESI following vaccination and that the vaccination of these vulnerable citizens should be prioritized.

This study has several strengths. First, the databases covered almost the entire population of Hong Kong, conferring high population representativeness. Second, a variety of sensitivity analysis confirmed the robustness of the results. Third, vaccine safety with regard to underlying multimorbidity, especially the safety profile of CoronaVac, was little investigated and this study provides preliminary evidence on this under-researched topic. Despite these strengths, there are several limitations to this study as well. First, we only had access to public healthcare databases and patients whose chronic conditions were managed in the private sectors were not included. There may be potential under-ascertainment in people only using private healthcare services as a result. However, since Hong Kong does not have a comprehensive publicly funded primary care system, the majority of chronic diseases are managed by the specialist outpatient clinics of the HA. Previous research has suggested that a vast majority of chronic disease patients in Hong Kong had typically used public services and the number of omitted patients should have limited impact on the results[35]. In addition, both inpatient ICD-9 and outpatient ICPC-2 were used to operationalize underlying conditions to minimize the omission of patients. Second, AESI may be handled in settings beyond public healthcare in the city, such as the private sector or overseas. Nevertheless, in terms of the number of hospital admissions that are warranted for most of the included AESI, the HA constitutes ~80% of the market share in Hong Kong[36]. Third, residual confounding such as the healthy user bias observed in the study is probably because the variety of covariates considered in the analysis may not be sufficiently wide subject to data availability. Specifically, the inclusion of socio-economic status, lifestyle factors and previous SARS-CoV-2 infection as covariates would be of great value to further research. Fourth, similar to other large-scale pharmacovigilance studies using electronic medical record databases, we only relied on the diagnostic codes and other records for the operationalization of the diseases despite the demonstrated accuracy of those codes[37], and specifically to distinguish between overlapping AESI and pre-existing chronic conditions. Last, the fact that the vulnerable populations living with chronic conditions are being vaccinated less proactively in Hong Kong compared with other societies may limit the generalizability of the findings. Also, as the population of Hong Kong is predominantly Chinese, replication of the analyses in other world populations is warranted.

In this post-marketing pharmacovigilance study of 883,416 individuals with chronic diseases, we found a low incidence of AESI and no evidence of a modified association between Covid-19 vaccination and AESI by multimorbidity status.

## Methods

**Study design**. We adopted a retrospective cohort study design to examine the association between vaccination and the risk of AESI 28-day post-vaccination as well as the effect modification by multimorbidity status.

**Data source**. De-identified electronic medical records of patients (aged 16 years or older) were provided by the HA, the sole provider of public inpatient services and a major provider of public outpatient services in Hong Kong. De-identified vaccination records provided by the Department of Health were linked by matching an encrypted unique person ID (Hong Kong Identity Card Number or foreign passport number) between the two databases. The Department of Health is the Government agency responsible for overseeing the implementation of mass vaccination in Hong Kong. Under the Prevention and Control of Disease (Use of Vaccines) Regulation (Cap 599K)[25], all Covid-19 vaccines administered were recorded with personal identification information in order to facilitate an efficient tracking of vaccine recipients in case of urgent safety issues. These two sources of data have been used for previous Covid-19 vaccine safety research[7,12]. Previous research validating the diagnostic codes of the HA data showed that for myocardial infarction and stroke, the positive predictive value was estimated above 85%[37]. The HA databases have previously been used extensively for drug safety and epidemiological studies[38–40]. It is a longitudinal electronic medical record database allowing us to identify the newly recorded diagnoses.

This study was approved by the institutional review board of the University of Hong Kong / Hospital Authority Hong Kong West Cluster (UW 21-149 and UW 21-138) and the Department of Health Ethics Committee (LM 21/2021).

**Cohort selection**. The mass Covid-19 vaccination program in Hong Kong was launched on 23 February 2021 for CoronaVac and 6 March 2021 for Comirnaty. The first batch of residents eligible for vaccination consisted of healthcare workers, those aged 60 or above (and carers of those aged 70 or above), staff of care homes, essential public service providers, and cross-boundary transportation or port workers. The second batch (March 8) were food handlers, public transport service workers, construction workers, school teachers and staff, and staff of the tourism industry. The third batch (March 16) further included people aged 30–59, students aged 16 or older studying abroad, as well as domestic helpers. Starting from April 15, other people aged 16 or above were made eligible. People were given the free choice of deciding whether or not to receive the vaccines, which one of the two vaccines to receive, as well as the venue of vaccination, but a different vaccine for the second dose was not allowed[23]. For Comirnaty, contraindications are hypersensitivity to previous doses of Comirnaty, or to the active substance or to any of the excipients. For CoronaVac, contraindications include a history of allergic reactions, severe underlying conditions, pregnancy, or lactation.

We retrieved the records of patients who received inpatient or outpatient services provided by the HA between 1 January 2018 and 31 July 2021, and selected those ever coded with a diagnosis of any of 20 chronic conditions in their medical records since 2018 based on a widely used list of conditions for multimorbidity operationalization[41] including hypertension, diabetes mellitus (type 2), severe constipation, depression, cancer, hypothyroidism, chronic pain, asthma, alcohol misuse, chronic pulmonary disease, schizophrenia, rheumatoid arthritis, peptic ulcer disease, cirrhosis, psoriasis, Parkinson's disease, dementia, irritable bowel syndrome, inflammatory bowel disease, and peripheral vascular disease using International Classification of Diseases, Ninth Revision (ICD-9) and International Classification of Primary Care, Second Edition (ICPC-2). Diseases that overlapped with the AESI investigated (based on ICD-9) were not considered. Supplementary Table 15 shows the ICD-9 and ICPC-2 codes used to identify the patients. Subsequently, age and sex were used to match the vaccinated individuals with unvaccinated individuals at the ratio of one to three with the first-dose vaccination date of vaccinated individuals mapped to the matched unvaccinated individuals as the index date (23 February 2021 onwards). Specifically, three randomly selected unvaccinated individuals of the exact same age and same sex were matched to each vaccinated individual. We further removed unvaccinated people who died before the index date, those who were hospitalized on the index date, those who had their first chronic disease diagnoses only after the index date, or those who had AESI records before the index date. We further removed those who died before the index date, were hospitalized on the index date, had their first chronic disease diagnoses only after the index date, or had AESI records before the index date. We retrospectively examined the medical records of the patients up to 2005 to avoid omitting any prior diagnosis of AESI.

**Outcome: adverse events of special interest**. We followed the World Health Organization's Global Advisory Committee on Vaccine Safety (GACVS)[42] and adopted a list of 30 AESI (please see Supplementary Table 16), to define the primary composite outcome of this study using both inpatient and outpatient diagnoses, i.e., time to any AESI from the index date. Observation also ended with 28 days after the index date, death, receiving the second dose, and 31 July 2021 (end of available data), whichever came earliest. The recommended dosing intervals (number of days between the two doses) were 21 for Comirnaty and 28 for CoronaVac[25]. Eight sub-categories of the AESI according to GACVS, namely, auto-immune diseases, cardiovascular

system diseases, circulatory system diseases, hepato-renal system diseases, nerves and central nervous system diseases, skin and mucous membrane, bone and joints system diseases, respiratory system diseases, and diseases of other systems, as well as each of the 30 specific AESI were used as the secondary outcomes.

**Exposure: receiving CoronaVac or receiving Comirnaty**. Receiving CoronaVac or receiving Comirnaty as compared with being unvaccinated were adopted as the exposure of this study. As patients are not allowed to switch between vaccine types because of the centralized booking system managed by the Hong Kong Government, these three categories were mutually exclusive.

**Effect modifier: multimorbidity**. Multimorbidity was dichotomized as being multimorbid (with two or more listed chronic conditions) versus only one condition prior to the index date.

**Multiple vaccination group weighting**. Similar to the inverse probability of treatment weighting method, we used entropy balancing[43] implemented by the R package 'WeightIt' to assign an optimized set of weights to the patients in the cohort to generate balanced cohorts considering the potential confounding effects of age, sex, and each of the 20 chronic conditions used to identify the cohort[43]. The SMD between the unvaccinated, CoronaVac and Comirnaty groups were examined with the maximum differences (among the three between-group differences) being smaller than 0.1 indicating a balance between the three groups[38].

**Statistical analysis**. We implemented a Cox proportional hazard model to examine the association between vaccination and AESI in the weighted cohort. Three models were constructed. First, we included the vaccine group only. Second, we further included multimorbidity to examine the association between multimorbidity and AESI. Third, we specified an interaction between vaccination and multimorbidity to test for the differences of the association of vaccination with AESI between multimorbid patients and those living with only one condition. The same analyses were replicated on all sub-categories of AESI as secondary outcomes. All statistical tests used in this study were two-sided tests. A P-value of 0.05 or below was considered indicative of statistical significance.

**Sensitivity analyses**. A series of sensitivity analyses were conducted to test for the robustness of the results. First, we replicated the main analysis with the 28th day following the index date and date of second dose omitted as observation endpoints. This analysis was to examine any potentially different results arising from including the observation of the second dose. Second, we replicated the analysis on only those who were vaccinated on or before 3 July 2021 to allow all patients to have at least 28 days of observation. Third, we replicated the main analysis on those with only ICD-9 diagnoses in our records, with multimorbidity status replaced by the Charlson Comorbidity Index scores[44] to take into consideration the severity of diseases. Fourth, we replicated the analysis with AESI outcomes defined by inpatient records only and excluded outpatient records to minimize misclassification of follow-up visits in the outpatient setting. Fifth, we replicated the analysis with patients with a hospitalization record within six months prior to the baseline removed. Sixth, we replaced the dichotomous multimorbidity status as an effect modifier with the number of listed chronic conditions, a continuous independent variable. Seventh, a replication of the analysis was conducted with Poisson regression to generate incidence rate ratios for each outcome. Eighth, we used propensity score-based generalized boosted model for an alternative weighting method for the analyses using the same R package, i.e. 'WeightIt'. Last, we replicated the age- and sex-matching with replacement and repeated the analysis with the resulted cohort.

All analyses were conducted using the R statistical environment (Version 4.1.1, Vienna, Austria). For the weighting of the cohort, R package 'WeighIt' was used and 'Survival' was used for Cox regression. In common with all large-scale electronic medical record database research, we relied on the medical records for the operationalization of diseases and AESI, and it was presumed there were no missing data in our statistical analyses.

**Reporting summary**. Further information on research design is available in the Nature Research Reporting Summary linked to this article.

## Data availability

Data are not available as the data custodians (the Hospital Authority and the Department of Health of Hong Kong SAR) have not given permission for sharing due to patient confidentiality and privacy concerns. Local academic institutions, government departments, or non-governmental organizations may apply for the access to data through the Hospital Authority's data sharing portal (https://www3.ha.org.hk/data).

## Code availability

The code used for this study is available on Zenodo (https://doi.org/10.5281/zenodo.5792703).

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

## Acknowledgements

This study was funded by a research grant from the Food and Health Bureau, Government of the Hong Kong Special Administrative Region of China to Prof. Ian Chi Kei Wong (reference COVID19F01). We thank Mr. Joseph Blais for language proofreading this manuscript.

## Author contributions

F.T.T.L. had the original idea for the study, constructed the study design and the analytic plan, and wrote the first draft of the manuscript. L.H. extracted data and performed statistical analysis. F.T.T.L. cross-checked the results. C.S.L.C., E.Y.F.W., X.L., C.K.H.W., E.W.W.C., T.M., H.L., E.W.Y.C., and I.C.K.W. provided critical input to the analyses, design, and discussion. D.H.L. assisted with the literature review. J.C.N.L. assisted with formatting the figures. E.Y.F.W., C.S.L.C., and I.C.K.W. have accessed and verified the data used in the study. I.C.K.W. is the principal investigator and provided oversight for all aspects of this project. All authors contributed to the interpretation of the analysis, critically reviewed, and revised the manuscript, and approved the final manuscript as submitted. All authors had full access to all the data in the study and had final responsibility for the decision to submit for publication.

## Competing interests

F.T.T.L. has been supported by the RGC Postdoctoral Fellowship under the Hong Kong Research Grants Council and has received research grants from the Food and Health Bureau of the Government of the Hong Kong Special Administrative Region, outside the submitted work. C.S.L.C. has received grants from the Food and Health Bureau of the Hong Kong Government, Hong Kong Research Grant Council, Hong Kong Innovation and Technology Commission, Pfizer, IQVIA, and Amgen; and personal fees from PrimeVigilance; outside the submitted work. E.Y.F.W. has received research grants from the Food and Health Bureau of the Government of the Hong Kong Special Administrative Region, and the Hong Kong Research Grants Council, outside the submitted work. X.L. has received research grants from the Food and Health Bureau of the Government of the Hong Kong Special Administrative Region; research and educational grants from Janssen and Pfizer; internal funding from the University of Hong Kong; and consultancy fees from Merck Sharp & Dohme, unrelated to this work. E.W.Y.C. reports honorarium from Hospital Authority; and grants from Research Grants Council (RGC, Hong Kong), Research Fund Secretariat of the Food and Health Bureau, National Natural Science Fund of China, Wellcome Trust, Bayer, Bristol-Myers Squibb, Pfizer, Janssen, Amgen, Takeda, and Narcotics Division of the Security Bureau of the Hong Kong Special Administrative Region, outside the submitted work. I.C.K.W. reports research funding outside the submitted work from Amgen, Bristol-Myers Squibb, Pfizer, Janssen, Bayer, GSK, Novartis, the Hong Kong Research Grants Council, the Food and Health Bureau of the Government of the Hong Kong Special Administrative Region, National Institute for Health Research in England, European Commission, and the National Health, and Medical Research Council in Australia; has received speaker fees from Janssen and Medice in the previous 3 years; and is an independent non-executive director of Jacobson Medical in Hong Kong. All other authors declare no competing interests.
