## [Peer Review File · Nature Communications]

Multimorbidity and adverse events of special interest associated with Covid-19 vaccines in Hong KongREVIEWER COMMENTS

Reviewer #1 (Remarks to the Author):

This study sought to investigate the additional risk of AESI within 28 days of the first vaccine dose of CoronaVac or Comirnaty in those with multi-morbidity. Using routinely available data, the authors constructed a population-based cohort study focusing on 30 AESI identified by WHO. The authors found no additional risk of these outcomes following vaccination in those with multi-morbidity.

Overall, the study appears reasonably well conducted. The big issue I struggled with however is why they thought there would be an increased risk in those with multi-morbidity. The rationale for this investigation is to my mind not adequately made. The Introduction needs to be strengthened in this respect.

Other limitations/areas of improvement include:

1. Bringing the literature review up-to-date as there are a number of important omissions in this rapidly evolving field of inquiry
2. More context would be helpful as to the criteria by which individuals received one or other vaccine
3. Whilst it is reasonable to use the WHO's list of AESIs, I note that CVST is not listed in Table e8
4. I am unclear if there is a risk of potential under-ascertainment for those managed in primary care settings
5. The inability to adjust for potential confounders such as socioeconomic status, smoking status, and SARS-CoV-2 infection is an important limitation
6. I would have liked to see an accompanying protocol and accompanying statistical analysis plan with details of any protocol deviations.

Reviewer #2 (Remarks to the Author):

I appreciated the opportunity to review this paper, which examined adverse events of special interest (AESI) after COVID-19 vaccination with CoronaVac or Comirnaty vaccines in Hong Kong, with special attention to whether multimorbidity impacted the rate of these AESI.

Data came from deidentified electronic medical records from the health authority which is the sole provider of public inpatient services and a major provider of public outpatient services in Hong Kong. This high coverage of health services in the EMR is a strength. The AESI were selected with reference to the World Health Organization's list of relevant events and categorized according to the WHO GACVS. In considering multimorbidity status, diseases which overlapped with the AESI were not considered, which is sensible. Multimorbidity was considered as a dichotomized variable and defined as two or more of the listed chronic conditions.

In the extraction of the data set, the authors only considered AESI after the first dose in the main analysis (though in the sensitivity analysis, they did look at longer follow up times to examine any potentially differing results following the second vaccine dose and no difference was identified). Weighting was done to ensure balanced groups; the weighting methodology was rigorous and resulted in successfully balanced the groups.

Records for over 800,000 patients were included, and 0.3% had AESI. Although patients with multimorbidity had an increased risk of events regardless of vaccination status (vs. those without multimorbidity), there was no evidence that multimorbidity was associated with extra risk of AESI following COVID-19 vaccination. Indeed, among patients with multimorbidity the risk of these events was lower in those who were vaccinated with either vaccine product vs. the unvaccinated group.

Several sensitivity analyses were conducted – this was another strength of the manuscript, and was reassuring in that they resulted in findings consistent with the interpretation of the results from the main analyses.

Overall, this is a well-written paper investigating a topic of high interest in the current context of pandemic vaccine programs. The methods are sound and well described and the paper makes a good contribution to the literature.

I note the following issues:

1. The authors state that the median follow-up times for the two vaccines were different at 28 (CoronaVac) and 21 (Comirnaty) days. This could be better explained. Is it because the rollout of the Comirnaty program was later and thus less time was available for follow up? Or is it because a second dose was then administered at 21 days making it difficult to attribute any issues to the first dose? This could be clarified in the manuscript.
2. The authors used a dichotomous multimorbidity variable. This is a reasonable and clear/simple approach, though it would be interesting to know whether treating multimorbidity as a continuous variable would have any impact on the results (though with the lack of any signal of concern it seems unlikely). To some degree this may have been addressed by the sensitivity analysis using the Charlson Comorbidity Index.
3. It is interesting to note that the finding of there being higher risk of adverse events in the multimorbidity group (regardless of vaccination status) (HR 1.63 in the main analyses) was accentuated (HR 1.76) when restricted to only the (presumably more severe) AESI from the sensitivity analysis restricted to inpatient records as shown in e Table 5, presumably reflecting that multimorbidity was even more predisposing to more severe events (and/or a higher baseline burden of illness leading to admission). It would be interesting to hear the authors' interpretation of this.
4. In the second paragraph of the Discussion, the authors state that the finding of a lower risk of AESI among patients receiving the vaccines could reflect an indication bias, where patients who decided to get vaccinated were those who have better control of their chronic conditions even with the same diagnosis. I wonder whether this is best called an indication bias or is it perhaps more of a healthy user bias?
5. Page 4, lines 100-102: this sentence is unclear and could be edited for clarity.
6. I did not see an ethics statement, which should generally be included in the manuscript (depending of course on the journal formatting guidelines).

Melissa K Andrew

Reviewer #3 (Remarks to the Author):

Key results

These are that the two vaccines used in Hong Kong did not result in a greater incidence, following a first dose of vaccine, of adverse events of special interest (AESI), as defined by WHO, in those who had multimorbidity compared with those who had a single morbid condition.

Validity

It is not clear that the data sources used will have truly captured the AESI, but there is no strong reason to doubt their validity. There are a number of questions around the data and analysis methods that should be answered (see notes below).

Significance

The particular question being answered in this manuscript has not been addressed in any detail before, as far as I am aware. It does not show that the overall risk-benefit balance of vaccination is favourable (which has been shown elsewhere) but does provide some reassurance in the population with multimorbidity.

Data and methodology

There are some questions about the data and methodology, which are given below.

Analytical approach

It is likely that the approach is satisfactory but some more details need to be given about the analysis.

Suggested improvements

A number of detailed suggestions are given below which might improve the submitted manuscript.

Clarity and context

Generally this is clear. The authors are constrained by the Nature guidelines which, in leaving the methods to the end of the paper, make it less clear what is being done in the study.

References

The field of Covid vaccines and their adverse effects and benefits is moving very rapidly and the authors may wish to bring them up-to-date. Some references quoted do not make the points that the authors suggest they do and should be deleted or replaced.

Further general and detailed points are given in the numbered lists below.

General Points

1 This paper could potentially be of considerable interest for two main reasons. Firstly it includes a description of rates of some adverse events following the Chinese vaccine CoronaVac which has had relatively little published on it, and secondly it is a review of whether multimorbidity is associated with increased adverse events as a result of vaccination.

There are several questions as to whether the paper as it stands fulfils these objectives.

2 The important overall questions are whether the basic data are adequate and reliable and whether the overall methods are appropriate. For example, is there absolute certainty that the vaccination records are reliable and totally unbiased? From the Johns Hopkins data (which itself may not be reliable), it seems there were very few cases of Covid occurring and there seems to be no discussion of the status of the epidemic in Hong Kong between February 23rd 2021 (when presumably analysis and follow-up time starts), to July 31st 2021 (when the follow-up was stopped). While this may mean that the benefits of vaccination are not able to be seen, it may mean the overall results are not easily generalisable.

3 An important issue relates to whether the use of all adverse events of special interest [AESI] as a single outcome is sensible. It also calls into question the use of a time-to-event analysis like the Cox model for these events as a whole. They have quite different likely patterns of occurrence over time, and the use of the first AESI may not reflect anything of real interest. For example, anaphylaxis would be expected to occur very rapidly after vaccination and virtually never in an unvaccinated group. Grouping that with anosmia which has a quite different pattern over time (as would type I diabetes, Bell's palsy and many of the other AESI) makes neither medical nor statistical sense. The seriousness of the different AESI is also very varied. Anosmia, while very frustrating is not life-threatening, while the cardiac and circulatory diseases could be life-threatening. The eight sub-categories, while having some related terms within a category, also have some which are only perhaps superficially related, e.g Multisystem inflammatory syndrome in children and anosmia.

Detailed points

1 (line) L43- 45: surely the safety profile of vaccines is relevant in all situations and it could be argued that it is especially in countries where SARS-CoV-2 infection rate is uncontrolled. Countries where vaccine coverage is low have the potential to be more affected by issues of vaccine safety than in those where most people are already vaccinated.

2 L46: The lists of adverse events of special interest [AESI] were initially developed prior to knowledge of the actual adverse events occurring in those vaccinated. Might the authors wish to group the AESI by whether they are known to be caused by SARS-CoV-2 infection, and by whether they are known to be caused by vaccination?

3. L49-51: Where is the evidence that there is increased concern over multimorbidity? There has been concern over a long period. Reference 8 (from 2012) also makes it clear there is a very strong association between age and multimorbidity such that it occurs in over 80% of those aged over 85 but only 2% in those aged under 25 and 11% in those 25-44. Is there a similar pattern seen in Hong Kong? Table 1 does not give a breakdown of multimorbidity by age but seems to suggest it is higher (37%) in the unvaccinated than in the vaccinated (<30%). This seems a strange pattern, though we do not have those with no morbidities in this study which makes comparison with reference 8 difficult. The WHO guidance, reflected in for example the Pan-American and US guidelines and in this paper, cited by WHO- [<https://journals.plos.org/ploscompbiol/article?id=10.1371/journal.pcbi.1008849>] and as far as I am aware, in virtually all countries recommended vaccination to those most vulnerable to serious health outcomes from Covid-19 disease. Was a different pattern pursued in Hong Kong? This point is also relevant at line 120 of the paper.

4 L56: it is not clear that these references suggest those with multimorbidity require study.

5 L67-68: It is not clear how close age-matching was? The exact method of doing the matching does need to be set out because matching in cohort studies can lead to bias if not done correctly. (Sampling without replacement can lead to the pool of available controls having an induced selection bias).

6 L65-72: The text here does not seem to have a logical flow, while Figure 1 is a little clearer. The object was to have 3 unvaccinated controls for each vaccinated index patient [L197], but it is not clear why the final ratio is 1:1.84 (see Figure 1) before exclusions and after exclusions is 1:1.63. This failure to reach the intended matching ratio has the potential to introduce bias.

7 L70: Why quote median length of follow-up? It would be clearer to have mean times since the distribution is close to rectangular with many tied observations so medians and IQRs are uninformative.

8 L88: there should be a hyphen after "Kaplan" and before "-Meier".

9 L91: "vaccination status" is misleading here; should it be "the two vaccine groups"?

10 L102-3: eTable 1 does show a number of differences in the overall pattern, particularly in the differences between the two vaccines in their point estimates for several factors, and in the variability of the effect of multi-morbidity on the different groups of AESI. As a minor point, in the last line of data in eTable 1 the interaction term- CoronaVac X multimorbidity, with an HR of 1.26e-08 (4.18e-10, 3.79e-07) *** may be what the computer output suggests but this is a spurious finding and suggests that the software has some numerical error. There are very few cases in the skin AESI group (as evidenced by the generally extremely wide confidence intervals for other estimates and shown in Table 1) and so this line is unreliable. It's possible that the skin grouping (Erythema multiforme & Chilblain – like lesions which is a strange pair of terms) is picking up vaccine reactogenicity rash, which will be caused by the vaccines rather than preventing them.

11 L105: I think the interaction that is "significant" is over-interpreted and possibly also mis-

interpreted. Surely it is a stronger effect of Comirnaty with multi-morbidity rather than a weaker effect? The interaction estimate of 0.72 is surely a multiplier of the HR and not an absolute value of the HR in those receiving Comirnaty? Given the lack of interaction overall should this even be mentioned?

12 L116-126: The arguments here may not be well-based. Reference 16 does not relate to control of chronic conditions, but notes the very well known bias- the "healthy vaccinee effect". Those who are feeling unwell delay vaccination until they feel better. Also, guidelines suggest delaying vaccine after a positive SARS-CoV-2 test for a few days. The main morbidities here are hypertension and diabetes and it is not clear that control of these conditions affects receipt of a vaccine. Usual guidelines would suggest that those who are chronically unhealthy should receive vaccination but if that is not the case in Hong Kong then again the results may not be generalisable.

13 L126: Might the phrase "If multimorbidity does imposes additional AESI risk increase following vaccination" be better as "If multimorbidity does impose additional AESI risk following vaccination"?

14 L134-140: This is a very good point and might be emphasised more- not sure if it should be in the Abstract?

15 L147: as far as this reviewer can tell, reference 21 says nothing about vaccine hesitancy or concerns on safety being greater in those with multi-morbidity. It is a paper about parental attitudes towards childhood vaccines and surely of limited relevance here?

16 L153-155: As noted above, this argument seems mistaken- where the infection is under control vaccine coverage is already likely to be high, but the findings are surely relevant whatever the level of control of the infection. This point does not need to be made.

17 L155-156: the "third point" is unclear. The overall coverage seems to be 41% from Figure 1, but this probably varies a great deal with age. Why is it important to say that since most people do not receive the vaccine the absolute risk of AESIs is low? This seems to not be a sensible point.

18 L179: What is the nature of the ID used for matching?

19 L188: How was information from 2005 to 2018 obtained? Is it that for those included between 2018 and 2021, their entire records going back to 2005 were obtained? When did the EHR system start? Has there been validation of the disease coding within the database over this period?

20 L194: It is obviously difficult to deal with medical events and conditions that could contribute to the multimorbidity but may also be classed as AESI if they occurred after vaccination. It is not clear whether this is a particular problem with diabetes and coronary artery disease. Type I diabetes is stated to be an AESI and type II is a morbidity. Will it always be possible to be clear about this? Is the coding in the EHRs sufficiently good to be sure that events that are recorded after vaccination are truly newly incident events? Hypertension is strongly associated with CAD; can the authors be sure the CAD are incident events? Some more detail of which aspects of morbidity were excluded might be given unless it is simply the codes in eTable 8.

21 L205: Should the paper as a whole emphasise that it is just looking at first doses of vaccine (Censoring takes place at the second dose)? The main text of the paper does not make this clear- it continually refers to "vaccination".

22 L211; should it be "Receiving CoronaVac or receiving Comirnaty"? A reader might think patients received both vaccines when none did.

23 L219-224: The use of entropy balancing is interesting, but it is unclear how with only binary covariates it differs from propensity score weighting? In addition, its usual use is with a binary treatment variable. Was the weighting method applied using vaccinated compared with unvaccinated? Did it involve a further step to compare the two vaccine groups? Was a conventional propensity score weighting used and if so what differences in results were obtained?

24 L226-227: the entropy-balancing method usually relies on using logistic regression for estimation of causal effects. While it probably makes little difference, technically the Cox model has untestable assumptions when used to estimate causal effects. As noted above, there are several reasons for using an alternative analysis model. It is not clear what is meant by "vaccination status- it implies it is a comparison of vaccinated with unvaccinated, but it seems that model 1 (and the other models) use three treatment groups. This should be clearer.

25 L243: There are very many possible packages able to be used for the analyses done here and to simply quote the "R statistical environment" is an inadequate description.

26 L244: Surely it is only assumed that there are no missing data? For each of the AESI for example, their presence is what is noted in the records, their absence is not actually noted but presumed.

27 Figure 1: the number of exclusions seems quite large and to have about 3% of the vaccinated patients getting a new chronic condition diagnosed in the 28 days after vaccination seems strange. The percentage seems a bit lower in the unvaccinated (possibly not unexpected) but still seems rather high. Is a comment necessary?

28 Figure 2: it is usual for adverse events, to plot their cumulative frequency or cumulative hazard rather than a Kaplan-Meier survival curve. The figure for the Comirnaty treated patients seems strange. The truncation of the curve for those without multimorbidity at 22 days and the steep rise in the number of AESIs after 24 days for those with multi-morbidity require some comment. The curves seem to be drawn smoothly but the data imply it should be a series of steps.

29 Figure 3. The chord diagrams are an appealing way of displaying the data, but it could be good to have some extra eTables giving actual numbers.

30 Table 1: I do not understand how the weighted numbers for the vaccinated groups are different from the unweighted? Surely the balancing is to make the unvaccinated similar to the vaccinated. Is it because the unweighted data is before matching which might have led to a lot of exclusions? Should there therefore be an extra column of the unweighted but matched data? This all seems rather puzzling.

Response to Reviewer Comments

Manuscript Title: *Multimorbidity and adverse events of special interest associated with CoronaVac (Sinovac) and Comirnaty (Pfizer-BioNTech)*

Journal: *Nature Communications*

Ref. No.: *NCOMMS-21-35429*

November 19, 2021

Dear Reviewers,

Thank you very much for reviewing this manuscript. We are very grateful for your insightful comments. Please find appended below our detailed response to each of your specific comments with proposed changes and additional analyses. The revised relevant text is quoted as appropriate:

Reviewer #1

1.1. “This study sought to investigate the additional risk of AESI within 28 days of the first vaccine dose of CoronaVac or Comirnaty in those with multi-morbidity. Using routinely available data, the authors constructed a population-based cohort study focusing on 30 AESI identified by WHO. The authors found no additional risk of these outcomes following vaccination in those with multi-morbidity.

Overall, the study appears reasonably well conducted. The big issue I struggled with however is why they thought there would be an increased risk in those with multi-morbidity. The rationale for this investigation is to my mind not adequately made. The Introduction needs to be strengthened in this respect.”

Author Response:

Thank you very much for the encouraging overall comment on our work. We have now specified the rationale for this investigation more explicitly that adverse drug reactions among people living with multimorbidity have been shown to be more common than those living without because of the complex mechanisms underlying the co-occurring conditions and co-medications. Also, once infected, these people are at much elevated risks of serious complications compared with those without chronic conditions.

“Previous research before the pandemic has shown a potential risk increase of cardiovascular events and other adverse outcomes in people living with multimorbidity compared with those without.^{16,17} Specifically, the complex underlying mechanisms of the co-occurrence of multiple health conditions and multiple medications have been found to be related to increased adverse drug reactions.¹⁸⁻²⁰ Of note, people living with multimorbidity are at increased risk of serious complications following an infection of SARS-CoV-2.²¹”

Response to Reviewer Comments

Manuscript Title: *Multimorbidity and adverse events of special interest associated with CoronaVac (Sinovac) and Comirnaty (Pfizer-BioNTech)*

Journal: *Nature Communications*

Ref. No.: *NCOMMS-21-35429*

(Lines 50-55, P. 3)

“Existing research comparing the relationship between vaccination and AESI across sub-populations with and without multimorbidity is limited, rendering the risk and benefit assessment for the vaccination of the multimorbid populations inconclusive.”

(Lines 56-58, P. 3)

- 1.2. “Other limitations/areas of improvement include:
1. Bringing the literature review up-to-date as there are a number of important omissions in this rapidly evolving field of inquiry”**

Author Response:

Thank you for this suggestion. We have repeated the search of the literature for safety signals with regard to the use of Covid-19 vaccines among individuals with underlying conditions using keywords including specific chronic conditions and multimorbidities on PubMed. The relevant works are now cited.

“There is also increased safety concern regarding the vaccination of people living with underlying chronic conditions¹⁰⁻¹³ and multimorbidity,¹⁴ commonly referred to as the co-occurrence of two or more chronic health conditions in an individual.¹⁵”

(Lines 48 -50, P. 3)

- 1.3. “2. More context would be helpful as to the criteria by which individuals received one or other vaccine”**

Author Response:

Thanks for this suggestion. We have now provided more contextual information in the revised text. Specifically, the eligibility and roll-out order for different groups in society have now been specified.

“The mass Covid-19 vaccination program in Hong Kong was launched on February 23, 2021 for CoronaVac and March 6, 2021 for Comirnaty. The first batch of residents eligible for vaccination consisted of healthcare workers, those aged 60 or above (and carers of those aged 70 or above), staff of care homes, essential public service providers, and cross-boundary transportation or port workers. The second batch (March 8) were food handlers, public transport service workers, construction workers, school

Response to Reviewer Comments

Manuscript Title: *Multimorbidity and adverse events of special interest associated with CoronaVac (Sinovac) and Comirnaty (Pfizer-BioNTech)*

Journal: *Nature Communications*

Ref. No.: *NCOMMS-21-35429*

teachers and staff, staff of the tourism industry. The third batch (March 16) further included people aged 30 – 59, students aged 16 or older studying abroad, as well as domestic helpers. Starting from April 15, other people aged 16 or above were made eligible. People were given the free choice of deciding whether or not to receive the vaccines, which one of the two vaccines to receive, as well as the venue of vaccination, but a different vaccine for the second dose was not allowed.²³ For Comirnaty, contraindications are hypersensitivity to previous doses of Comirnaty, or to the active substance or to any of the excipients. For CoronaVac, contraindications include a history of allergic reactions, severe underlying conditions, pregnancy, or lactation.”

(Lines 218-230, P. 7)

1.4. “3. Whilst it is reasonable to use the WHO's list of AESIs, I note that CVST is not listed in Table e8”

Author Response:

Thank you for this comment. ICD-9 codes 325.0 and 437.6, commonly used to code CVST, were both included under thromboembolism (under circulatory system diseases) which was one of the 30 AESI examined in this study. Our analysis should have covered this important condition. Also, in the literature, so far only the AstraZeneca-Oxford Covid-19 vaccine (unavailable in Hong Kong), but not other vaccines, has been found associated with this particular condition.

1.5. “4. I am unclear if there is a risk of potential under-ascertainment for those managed in primary care settings”

Author Response:

Thank you for pointing this out. This potential under-ascertainment in the primary care setting is now acknowledged as a limitation to the study.

“First, we only had access to public healthcare databases and patients whose chronic conditions were managed in the private sectors were not included. There may be potential under-ascertainment in people only using private healthcare services as a result. However, since Hong Kong does not have a comprehensive publicly funded primary care system, the majority of chronic diseases are managed by the specialist outpatient clinics of the HA. Previous research has suggested that a vast majority of chronic disease patients in Hong Kong had typically used public services and the number of omitted patients should have limited impact on the results.³⁵ In addition, both

Response to Reviewer Comments

Manuscript Title: *Multimorbidity and adverse events of special interest associated with CoronaVac (Sinovac) and Comirnaty (Pfizer-BioNTech)*

Journal: *Nature Communications*

Ref. No.: *NCOMMS-21-35429*

inpatient ICD-9 and outpatient ICPC-2 were used to operationalize underlying conditions to minimize the omission of patients.”

(Lines 173-180, P. 6)

1.6. “5. The inability to adjust for potential confounders such as socioeconomic status, smoking status, and SARS-CoV-2 infection is an important limitation”

Author Response:

Thank you for this comment. We have now specified potential confounders which could have strengthened the analyses in the limitations section and recommended further research to include those.

“Third, residual confounding such as the healthy user bias observed in the study is probable because the variety of covariates considered in the analysis may not be sufficiently wide subject to data availability. Specifically, the inclusion of socioeconomic status, lifestyle factors and previous SARS-CoV-2 infection as covariates would be of great value to further research.”

(Lines 183-187, P. 6)

1.7. “6. I would have liked to see an accompanying protocol and accompanying statistical analysis plan with details of any protocol deviations.”

Author Response:

Thank you for this suggestion. This is a regulatory pharmacovigilance study set up by the regulatory authority to monitor the safety of vaccines. Unlike traditional academic research for which grant application and protocol were development in advance, we were responding to the local need and developed the study protocol in a timely manner; hence, we have no published protocol.

Nevertheless, we developed a brief unpublished analytic plan which is now attached to this submission. There are three differences between the actual analyses and those described in the plan. First, we included all Hospital Authority service users rather than only discharged inpatients as stated in the plan to confer stronger statistical power and include a more diverse mix of participants. Second, we specified we would use mixed-effects Cox regression to run the analyses to adjust for random effects across different hospitals but we were eventually not provided the data on the specific hospital in which the patients received care and decided to use fixed-effects Cox models accordingly.

Response to Reviewer Comments

Manuscript Title: *Multimorbidity and adverse events of special interest associated with CoronaVac (Sinovac) and Comirnaty (Pfizer-BioNTech)*

Journal: *Nature Communications*

Ref. No.: *NCOMMS-21-35429*

Third, we eventually used multiple group weighting (entropy rebalancing) instead of separate matching to adjust for covariates so that the comparison could be made simultaneously in one cohort. Last, the protocol specified the operationalization of multimorbidity status to be categorized into several groups, but we eventually had it dichotomized to present a clearer picture. Otherwise, the sample size for each group would be very small and results would become uninterpretable.

In the revised manuscript, we have added another sensitivity analysis with the dichotomized multimorbidity status as an effect modifier replaced by the number of chronic conditions to check if any discrepancies in the results arise. We observed no marked differences from the main analysis.

“Sixth, we replaced the dichotomous multimorbidity status as an effect modifier with the number of listed chronic conditions, a continuous independent variable.”

(Lines 296-298, P. 8)

Reviewer #2

2.1. “I appreciated the opportunity to review this paper, which examined adverse events of special interest (AESI) after COVID-19 vaccination with CoronaVac or Comirnaty vaccines in Hong Kong, with special attention to whether multimorbidity impacted the rate of these AESI.

Data came from deidentified electronic medical records from the health authority which is the sole provider of public inpatient services and a major provider of public outpatient services in Hong Kong. This high coverage of health services in the EMR is a strength. The AESI were selected with reference to the World Health Organization’s list of relevant events and categorized according to the WHO GACVS. In considering multimorbidity status, diseases which overlapped with the AESI were not considered, which is sensible. Multimorbidity was considered as a dichotomized variable and defined as two or more of the listed chronic conditions.

In the extraction of the data set, the authors only considered AESI after the first dose in the main analysis (though in the sensitivity analysis, they did look at longer follow up times to examine any potentially differing results following the second vaccine dose and no difference was identified). Weighting was done to ensure balanced groups; the weighting methodology was rigorous and resulted in successfully balanced the groups.

Records for over 800,000 patients were included, and 0.3% had AESI. Although patients with multimorbidity had an increased risk of events

Response to Reviewer Comments

Manuscript Title: *Multimorbidity and adverse events of special interest associated with CoronaVac (Sinovac) and Comirnaty (Pfizer-BioNTech)*

Journal: *Nature Communications*

Ref. No.: *NCOMMS-21-35429*

regardless of vaccination status (vs. those without multimorbidity), there was no evidence that multimorbidity was associated with extra risk of AESI following COVID-19 vaccination. Indeed, among patients with multimorbidity the risk of these events was lower in those who were vaccinated with either vaccine product vs. the unvaccinated group.

Several sensitivity analyses were conducted – this was another strength of the manuscript, and was reassuring in that they resulted in findings consistent with the interpretation of the results from the main analyses.

Overall, this is a well-written paper investigating a topic of high interest in the current context of pandemic vaccine programs. The methods are sound and well described and the paper makes a good contribution to the literature.”

Author Response:

Thank you very much for the positive overall comment on our manuscript.

- 2.2. “1. The authors state that the median follow-up times for the two vaccines were different at 28 (CoronaVac) and 21 (Comirnaty) days. This could be better explained. Is it because the rollout of the Comirnaty program was later and thus less time was available for follow up? Or is it because a second dose was then administered at 21 days making it difficult to attribute any issues to the first dose? This could be clarified in the manuscript.”**

Author Response:

Thank you for pointing this out. The recommended dosing interval (number of days between doses) was 21 days for Comirnaty and 28 days for CoronaVac. Therefore, you are right that typically at 21 days the observation for the Comirnaty group ended because of the second dose. We have now specified this dosing interval in the Methods section.

“The recommended dosing intervals (number of days between the two doses) were 21 for Comirnaty and 28 for CoronaVac.²⁴”

(Lines 256-257, P. 7)

- 2.3. “2. The authors used a dichotomous multimorbidity variable. This is a reasonable and clear/simple approach, though it would be interesting to know whether treating multimorbidity as a continuous variable would have any impact on the results (though with the lack of any signal of**

Response to Reviewer Comments

Manuscript Title: *Multimorbidity and adverse events of special interest associated with CoronaVac (Sinovac) and Comirnaty (Pfizer-BioNTech)*

Journal: *Nature Communications*

Ref. No.: *NCOMMS-21-35429*

concern it seems unlikely). To some degree this may have been addressed by the sensitivity analysis using the Charlson Comorbidity Index.”

Author Response:

Thank you for this helpful suggestion. We have now added another sensitivity analysis with the dichotomized multimorbidity status as an effect modifier replaced by the number of chronic conditions to check if any discrepancies in the results arise. We observed no marked differences from the main analysis.

“Sixth, we replaced the dichotomous multimorbidity status as an effect modifier with the number of listed chronic conditions, a continuous independent variable.”

(Lines 296-298, P. 8)

- 2.4. “3. It is interesting to note that the finding of there being higher risk of adverse events in the multimorbidity group (regardless of vaccination status) (HR 1.63 in the main analyses) was accentuated (HR 1.76) when restricted to only the (presumably more severe) AESI from the sensitivity analysis restricted to inpatient records as shown in e Table 5, presumably reflecting that multimorbidity was even more predisposing to more severe events (and/or a higher baseline burden of illness leading to admission). It would be interesting to hear the authors’ interpretation of this.”**

Author Response:

Thank you for pointing out this interesting finding. However, the confidence intervals overlapped and differences were non-significant. We have now inserted a brief interpretation of this contrast in the Discussion.

“Interestingly, we observed when the analysis was confined to AESI recorded in the inpatient setting, the increase of AESI risks associated with multimorbidity regardless of vaccination was greater although the confidence intervals overlapped and differences were non-significant. This may be because those AESI requiring tertiary care were more severe than the more broadly defined ones.”

(Lines 153-156, P. 5)

- 2.5. “4. In the second paragraph of the Discussion, the authors state that the finding of a lower risk of AESI among patients receiving the vaccines could reflect an indication bias, where patients who decided to get vaccinated**

Response to Reviewer Comments

Manuscript Title: *Multimorbidity and adverse events of special interest associated with CoronaVac (Sinovac) and Comirnaty (Pfizer-BioNTech)*

Journal: *Nature Communications*

Ref. No.: *NCOMMS-21-35429*

were those who have better control of their chronic conditions even with the same diagnosis. I wonder whether this is best called an indication bias or is it perhaps more of a healthy user bias?”

Author Response:

Thank you for this suggestion. We agree that ‘healthy-user bias’ better captures the essence of the phenomenon, and we have now used it throughout the text.

“This finding may reflect a healthy user bias whereby patients who decided to get vaccinated were those who had their chronic conditions better controlled even given the same diagnoses.”²⁵

(Lines 127-1128, P. 5)

“Third, residual confounding such as the healthy user bias observed in the study is probable because the variety of covariates considered in the analysis may not be sufficiently wide subject to data availability. Specifically, the inclusion of socioeconomic status, lifestyle factors and previous SARS-CoV-2 infection as covariates would be of great value to further research.”

(Lines 183-187, P. 6)

2.6. “5. Page 4, lines 100-102: this sentence is unclear and could be edited for clarity.”

Author Response:

Thank you very much. Please find the sentence rephrased for better clarity.

“Model 3 suggested there was no significant modification of AESI risk associated with vaccine group by multimorbidity (HR = 0.88, 95% CI 0.67 – 1.15 for Comirnaty; HR = 1.03, 95% CI 0.84 – 1.27 for CoronaVac).”

(Lines 106-108, P. 4)

2.7. “6. I did not see an ethics statement, which should generally be included in the manuscript (depending of course on the journal formatting guidelines).”

Author Response:

Thank you very much, we have included the name of the ethics committees with reference numbers in the Methods section.

Response to Reviewer Comments

Manuscript Title: *Multimorbidity and adverse events of special interest associated with CoronaVac (Sinovac) and Comirnaty (Pfizer-BioNTech)*

Journal: *Nature Communications*

Ref. No.: *NCOMMS-21-35429*

“This study was approved by the institutional review board of the University of Hong Kong / Hospital Authority Hong Kong West Cluster (UW 21-149 and UW 21-138) and the Department of Health Ethics Committee (LM 21/2021).”

(Lines 214-216, P. 7)

Reviewer #3

3.1. “Key results

These are that the two vaccines used in Hong Kong did not result in a greater incidence, following a first dose of vaccine, of adverse events of special interest (AESI), as defined by WHO, in those who had multimorbidity compared with those who had a single morbid condition.

Validity

It is not clear that the data sources used will have truly captured the AESI, but there is no strong reason to doubt their validity. There are a number of questions around the data and analysis methods that should be answered (see notes below).

Significance

The particular question being answered in this manuscript has not been addressed in any detail before, as far as I am aware. It does not show that the overall risk-benefit balance of vaccination is favourable (which has been shown elsewhere) but does provide some reassurance in the population with multimorbidity.

Data and methodology

There are some questions about the data and methodology, which are given below.

Analytical approach

It is likely that the approach is satisfactory but some more details need to be given about the analysis.

Suggested improvements

A number of detailed suggestions are given below which might improve the submitted manuscript.

Clarity and context

Generally this is clear. The authors are constrained by the Nature guidelines which, in leaving the methods to the end of the paper, make it less clear what is being done in the study.

References

The field of Covid vaccines and their adverse effects and benefits is moving very rapidly and the authors may wish to bring them up-to-date. Some references quoted do not make the points that the authors suggest they do and should be deleted or replaced.

Response to Reviewer Comments

Manuscript Title: *Multimorbidity and adverse events of special interest associated with CoronaVac (Sinovac) and Comirnaty (Pfizer-BioNTech)*

Journal: *Nature Communications*

Ref. No.: *NCOMMS-21-35429*

Further general and detailed points are given in the numbered lists below.”

Author Response:

Thank you very much for reviewing this manuscript thoroughly. Please find our detailed response to each of your comments appended below.

- 3.2. “1 This paper could potentially be of considerable interest for two main reasons. Firstly it includes a description of rates of some adverse events following the Chinese vaccine CoronaVac which has had relatively little published on it, and secondly it is a review of whether multimorbidity is associated with increased adverse events as a result of vaccination. There are several questions as to whether the paper as it stands fulfils these objectives.”**

Author Response:

Thank you for recognizing the strengths of this study. We have now included these under the strengths and limitations section in the revised manuscript.

“This study has several strengths. First, the low number of cumulative cases of Covid-19 (<12,500 out of over 7 million residents)³⁴ in the population facilitated the observation of the safety of vaccines with minimal impact from Covid-19 and its complications. Second, the databases covered almost the entire population of Hong Kong, conferring high population representativeness. Third, a variety of sensitivity analysis confirm the robustness of the results. Fourth, vaccine safety with regard to underlying multimorbidity, especially the safety profile of CoronaVac, was little investigated in the literature and this study addressed a novel research question.”

(Lines 166-172, P. 5-6)

- 3.3. “2 The important overall questions are whether the basic data are adequate and reliable and whether the overall methods are appropriate. For example, is there absolute certainty that the vaccination records are reliable and totally unbiased? From the Johns Hopkins data (which itself may not be reliable), it seems there were very few cases of Covid occurring and there seems to be no discussion of the status of the epidemic in Hong Kong between February 23rd 2021 (when presumably analysis and follow-up time starts), to July 31st 2021 (when the follow-up was stopped). While this**

Response to Reviewer Comments

Manuscript Title: *Multimorbidity and adverse events of special interest associated with CoronaVac (Sinovac) and Comirnaty (Pfizer-BioNTech)*

Journal: *Nature Communications*

Ref. No.: *NCOMMS-21-35429*

may mean that the benefits of vaccination are not able to be seen, it may mean the overall results are not easily generalisable.”

Author Response:

Thank you for this comment on the reliability of the databases. Regarding the vaccination records, we are rather confident they are reliable because Department of Health, the data provider, is the Government agency responsible for overseeing the implementation of mass vaccination in Hong Kong. Under the urgently enacted Prevention and Control of Disease (Use of Vaccines) Regulation (Cap 599K), all Covid-19 vaccines administered are recorded with personal identification information in order to facilitate an efficient tracking of vaccine recipients in case of urgent safety issues. We have now included this fact to support the reliability of the vaccination database.

“De-identified vaccination records provided by the Department of Health were linked by matching an encrypted unique person ID (Hong Kong Identity Card Number or foreign passport number) between the two databases. The Department of Health is the Government agency responsible for overseeing the implementation of mass vaccination in Hong Kong. Under the Prevention and Control of Disease (Use of Vaccines) Regulation (Cap 599K),²⁴ all Covid-19 vaccines administered are recorded with personal identification information in order to facilitate an efficient tracking of vaccine recipients in case of urgent safety issues.”

(Lines 203-209, P. 6)

We have now provided more contextual information about the epidemic in Hong Kong. It is indeed accurate that with few cases of Covid-19 in Hong Kong, the benefits of vaccination cannot be quantified. However, the focus of this investigation was the safety of vaccines and this low infection rate actually facilitates an ideal testing ground because the impact of Covid-19 and the associated complications on adverse events of special interest, if any, would be minimal and any safety signal would be likely be only related to the vaccine but not the epidemic. This has now been discussed as one of the strengths of the study too.

“This study has several strengths. First, the low number of cumulative cases of Covid-19 (<12,500 out of over 7 million residents)³⁴ in the population facilitated the observation of the safety of vaccines with minimal impact from Covid-19 and its complications. Second, the databases covered almost the entire population of Hong Kong, conferring high population

Response to Reviewer Comments

Manuscript Title: *Multimorbidity and adverse events of special interest associated with CoronaVac (Sinovac) and Comirnaty (Pfizer-BioNTech)*

Journal: *Nature Communications*

Ref. No.: *NCOMMS-21-35429*

representativeness. Third, a variety of sensitivity analysis confirm the robustness of the results. Fourth, vaccine safety with regard to underlying multimorbidity, especially the safety profile of CoronaVac, was little investigated in the literature and this study addressed a novel research question.”

(Lines 166-172, P. 5-6)

- 3.4. “3 An important issue relates to whether the use of all adverse events of special interest [AESI] as a single outcome is sensible. It also calls into question the use of a time-to-event analysis like the Cox model for these events as a whole. They have quite different likely patterns of occurrence over time, and the use of the first AESI may not reflect anything of real interest. For example, anaphylaxis would be expected to occur very rapidly after vaccination and virtually never in an unvaccinated group. Grouping that with anosmia which has a quite different pattern over time (as would type I diabetes, Bell’s palsy and many of the other AESI) makes neither medical nor statistical sense. The seriousness of the different AESI is also very varied. Anosmia, while very frustrating is not life-threatening, while the cardiac and circulatory diseases could be life-threatening. The eight sub-categories, while having some related terms within a category, also have some which are only perhaps superficially related, e.g Multisystem inflammatory syndrome in children and anosmia.”**

Author Response:

Thank you very much for this comment on the use of composite outcomes and time-to-event analysis. We have now replicated this analysis on all the specific AESI as additional secondary outcomes to see if any deviation from the main results arises. We also replicated the analysis using Poisson regression for the estimation of the incidence rate ratio without using a time-to-event outcome. Results all remained largely similar to the main analysis.

“Eight sub-categories of the AESI according to GACVS, namely, auto-immune diseases, cardiovascular system diseases, circulatory system diseases, hepato-renal system diseases, nerves and central nervous system diseases, skin and mucous membrane, bone and joints system diseases, respiratory system diseases, and diseases of other systems, as well as each of the 30 specific AESI were used as the secondary outcomes.”

(Lines 257-261, P. 7-8)

Response to Reviewer Comments

Manuscript Title: *Multimorbidity and adverse events of special interest associated with CoronaVac (Sinovac) and Comirnaty (Pfizer-BioNTech)*

Journal: *Nature Communications*

Ref. No.: *NCOMMS-21-35429*

“eTable 5 shows the results from Model 3 for all specific AESI. Most results were in line with the main findings with a few exceptions arising from very rare incidence (typically less than two observed cases for each group as shown in eTable 3), such as subacute thyroiditis, microangiopathy, erythema multiforme, anosmia, and ageusia.”

(Lines 112-115, P. 4)

“Seventh, a replication of the analysis was conducted with Poisson regression to generate incidence rate ratios for each outcome.”

(Lines 298-299, P. 8-9)

3.5. “Detailed points

1 (line) L43- 45: surely the safety profile of vaccines is relevant in all situations and it could be argued that it is especially in countries where SARS-CoV-2 infection rate is uncontrolled. Countries where vaccine coverage is low have the potential to be more affected by issues of vaccine safety than in those where most people are already vaccinated.”

Author Response:

Thank you for this comment. We agree that the safety profile of Covid-19 vaccines is important in all situations. We have now removed this phrase to avoid confusion.

“The safety of Covid-19 vaccines is of great public health concern and is crucial to tackling vaccine hesitancy amidst the pandemic.”

(Lines 43-44, P. 3)

3.6. “2 L46: The lists of adverse events of special interest [AESI] were initially developed prior to knowledge of the actual adverse events occurring in those vaccinated. Might the authors wish to group the AESI by whether they are known to be caused by SARS-CoV-2 infection, and by whether they are known to be caused by vaccination?”

Author Response:

Thank you for this suggestion, as evidence on the adverse outcomes following an infection of SARS-CoV-2 as well as the adverse event profiles of various vaccine platforms is still accruing, we believe it is too early to categorize AESI into infection-induced and vaccination-induced and decided to not conduct this categorization at this stage.

Response to Reviewer Comments

Manuscript Title: *Multimorbidity and adverse events of special interest associated with CoronaVac (Sinovac) and Comirnaty (Pfizer-BioNTech)*

Journal: *Nature Communications*

Ref. No.: *NCOMMS-21-35429*

- 3.7. **“3. L49-51: Where is the evidence that there is increased concern over multimorbidity? There has been concern over a long period. Reference 8 (from 2012) also makes it clear there is a very strong association between age and multimorbidity such that it occurs in over 80% of those aged over 85 but only 2% in those aged under 25 and 11% in those 25-44. Is there a similar pattern seen in Hong Kong? Table 1 does not give a breakdown of multimorbidity by age but seems to suggest it is higher (37%) in the unvaccinated than in the vaccinated (<30%). This seems a strange pattern, though we do not have those with no morbidities in this study which makes comparison with reference 8 difficult. The WHO guidance, reflected in for example the Pan-American and US guidelines and in this paper, cited by WHO- [https://journals.plos.org/ploscompbiol/article?id=10.1371/journal.pcbi.1008849] and as far as I am aware, in virtually all countries recommended vaccination to those most vulnerable to serious health outcomes from Covid-19 disease. Was a different pattern pursued in Hong Kong? This point is also relevant at line 120 of the paper.”**

Author Response:

Thank you very much for this comment. We have now provided more information on the roll-out plan in Hong Kong. The situation in Hong Kong is slightly different from many other parts of the world. Similar to other countries, the Hong Kong Government recommends high risk groups including those aged 60 or above to receive early vaccination; however, they also advised people with chronic conditions, especially those with their disease less well-controlled should seek their doctors' advice before they decide to receive the vaccine. Consequently, the older people and those living with chronic conditions show a higher degree of vaccine hesitancy. In fact, the vaccination rate among the age group of 80+ is as low as 16.9% as of 9 November 2021. This fact has now been inserted in the Discussion in the revised manuscript. We have also cited a more relevant commentary expressing concerns over the vaccination of people with multimorbidity.

“The mass Covid-19 vaccination program in Hong Kong was launched on February 23, 2021 for CoronaVac and March 6, 2021 for Comirnaty. The first batch of residents eligible for vaccination consisted of healthcare workers, those aged 60 or above (and carers of those aged 70 or above), staff of care homes, essential public service providers, and cross-boundary transportation or port workers. The second batch (March 8) were food

Response to Reviewer Comments

Manuscript Title: *Multimorbidity and adverse events of special interest associated with CoronaVac (Sinovac) and Comirnaty (Pfizer-BioNTech)*

Journal: *Nature Communications*

Ref. No.: *NCOMMS-21-35429*

handlers, public transport service workers, construction workers, school teachers and staff, staff of the tourism industry. The third batch (March 16) further included people aged 30 – 59, students aged 16 or older studying abroad, as well as domestic helpers. Starting from April 15, other people aged 16 or above were made eligible. People were given the free choice of deciding whether or not to receive the vaccines, which one of the two vaccines to receive, as well as the venue of vaccination, but a different vaccine for the second dose was not allowed.²³ For Comirnaty, contraindications are hypersensitivity to previous doses of Comirnaty, or to the active substance or to any of the excipients. For CoronaVac, contraindications include a history of allergic reactions, severe underlying conditions, pregnancy, or lactation.”

(Lines 218-230, P. 7)

“This observation is in line with the official guidelines published by the Hong Kong Government only recommending that patients living with chronic conditions receive the vaccine if their conditions are under stable control.²⁶ Our results may not be applicable to recipients who had poorly controlled chronic conditions. In fact, the vaccination rate among the age group of ≥ 80 was as low as 16.9% as of November 9, 2021.²⁷”

(Lines 128-132, P. 5)

“There is also increased safety concern regarding the vaccination of people living with underlying chronic conditions¹⁰⁻¹³ and multimorbidity,¹⁴ commonly referred to as the co-occurrence of two or more chronic health conditions in an individual.¹⁵”

(Lines 48 -50, P. 3)

We have now also inserted a frequency table showing the multimorbidity status by age and vaccine groups.

“eTable 1 shows the frequencies of patients by age and multimorbidity status. Unsurprisingly, multimorbidity was shown to be more prevalent among older age groups.”

(Lines 85-87, P. 4)

3.8. “4 L56: it is not clear that these references suggest those with multimorbidity require study.”

Author Response:

Response to Reviewer Comments

Manuscript Title: *Multimorbidity and adverse events of special interest associated with CoronaVac (Sinovac) and Comirnaty (Pfizer-BioNTech)*

Journal: *Nature Communications*

Ref. No.: *NCOMMS-21-35429*

Thank you for the helpful comments for clarification. We have now cited more relevant previous works on the elevated risk of observing adverse drug reactions among individuals living with multimorbidity than those living without.

“Specifically, the complex underlying mechanisms of the co-occurrence of multiple health conditions and multiple medications have been found to be related to increased adverse drug reactions.”¹⁸⁻²⁰

(Lines 52-54, P. 3)

3.9. “5 L67-68: It is not clear how close age-matching was? The exact method of doing the matching does need to be set out because matching in cohort studies can lead to bias if not done correctly. (Sampling without replacement can lead to the pool of available controls having an induced selection bias).

6 L65-72: The text here does not seem to have a logical flow, while Figure 1 is a little clearer. The object was to have 3 unvaccinated controls for each vaccinated index patient [L197], but it is not clear why the final ratio is 1:1.84 (see Figure 1) before exclusions and after exclusions is 1:1.63. This failure to reach the intended matching ratio has the potential to introduce bias.”

Author Response:

Thank you very much for pointing this out. We have now described the exact way we did the age- and sex-matching in the revised text. Specifically, the exact same age (in years) was required for the matching without replacement. Also, in order to check for any impact of the potential bias induced by this failure to match at the pre-specified ratio of 1:3, we have now conducted a sensitivity analysis whereby matching with replacement was used. Conclusions from the findings remained consistent with the main analysis. The Methods section has now been revised.

“Subsequently, age and sex were used to match the vaccinated individuals with unvaccinated individuals at the ratio of one to three with the first-dose vaccination date of vaccinated individuals mapped to the matched unvaccinated individuals as the index date (February 23, 2021 onwards). Specifically, three randomly selected unvaccinated individuals of the exact same age and same sex were matched to each vaccinated individual.”

(Lines 240-244, P. 7)

Response to Reviewer Comments

Manuscript Title: *Multimorbidity and adverse events of special interest associated with CoronaVac (Sinovac) and Comirnaty (Pfizer-BioNTech)*

Journal: *Nature Communications*

Ref. No.: *NCOMMS-21-35429*

“Last, we replicated the age- and sex-matching with replacement and repeated the analysis with the resulted cohort.”

(Lines 301-302, P. 9)

- 3.10. “7 L70: Why quote median length of follow-up? It would be clearer to have mean times since the distribution is close to rectangular with many tied observations so medians and IQRs are uninformative.”**

Author Response:

Thank you for this suggestion. Please find the summary statistics replaced by mean and standard deviation.

“The mean follow-up time [standard deviation (SD)] for the CoronaVac (n =182,442), Comirnaty (n =153,178), and unvaccinated groups (n =547,796) were 23.60 (8.17), 19.14 (6.57), 23.14 (8.45) days respectively.”

(Lines 72-74, P. 3)

- 3.11. “8 L88: there should be a hyphen after “Kaplan” and before “-Meier”.”**

Author Response:

Thank you. We have now replaced the Kaplan-Meier curves with cumulative incidence plots.

“Figure 2 shows the cumulative incidence of AESI by multimorbidity status and vaccine group over the follow-up period. Patients with multimorbidity were observed to have a faster increase of AESI incidence but no marked differences were identified between vaccine groups.”

(Lines 93-95, P. 4)

- 3.12. “9 L91: “vaccination status” is misleading here; should it be “the two vaccine groups”?”**

Author Response:

Thank you for the helpful comments for clarification. Please find the wording changed to avoid confusion.

Response to Reviewer Comments

Manuscript Title: *Multimorbidity and adverse events of special interest associated with CoronaVac (Sinovac) and Comirnaty (Pfizer-BioNTech)*

Journal: *Nature Communications*

Ref. No.: *NCOMMS-21-35429*

“Figure 3 shows three chord diagrams by vaccine group exemplifying the relative frequencies (represented by ribbon area) of AESI-chronic condition pairings with each color representing a specific AESI.”

(Lines 95-97, P. 4)

- 3.13. “10 L102-3: eTable 1 does show a number of differences in the overall pattern, particularly in the differences between the two vaccines in their point estimates for several factors, and in the variability of the effect of multi-morbidity on the different groups of AESI. As a minor point, in the last line of data in eTable 1 the interaction term- CoronaVac X multimorbidity, with an HR of 1.26e-08 (4.18e-10, 3.79e-07) *** may be what the computer output suggests but this is a spurious finding and suggests that the software has some numerical error. There are very few cases in the skin AESI group (as evidenced by the generally extremely wide confidence intervals for other estimates and shown in Table 1) and so this line is unreliable. It’s possible that the skin grouping (Erythema multiforme & Chilblain – like lesions which is a strange pair of terms) is picking up vaccine reactogenicity rash, which will be caused by the vaccines rather than preventing them.”**

Author Response:

Thank you for pointing out this observation. We have now rephrased and toned down the similarity between analyses on the primary and secondary outcomes in the revised text. We agree that the result for skin-related AESI maybe underpowered and refrained from highlighting the result or interpreting it.

“For analyses on sub-categories of AESI, results were largely similar with the main findings with differing degrees of variation of the point estimates (eTable 4). A significant positive interaction between multimorbidity and CoronaVac for skin, bone, and joints system AESI was found but it was only based on extremely few cases (eTable 3). eTable 5 shows the results from Model 3 for all specific AESI. Most results were in line with the main findings with a few exceptions arising from very rare incidence (typically less than two observed cases for each group as shown in eTable 3), such as subacute thyroiditis, microangiopathy, erythema multiforme, anosmia, and ageusia.”

(Lines 108-115, P. 4)

- 3.14. “11 L105: I think the interaction that is “significant” is over-interpreted and possibly also mis-interpreted. Surely it is a stronger effect of Comirnaty with multi-morbidity rather than a weaker effect? The**

Response to Reviewer Comments

Manuscript Title: *Multimorbidity and adverse events of special interest associated with CoronaVac (Sinovac) and Comirnaty (Pfizer-BioNTech)*

Journal: *Nature Communications*

Ref. No.: *NCOMMS-21-35429*

interaction estimate of 0.72 is surely a multiplier of the HR and not an absolute value of the HR in those receiving Comirnaty? Given the lack of interaction overall should this even be mentioned?"

Author Response:

Thank you very much for this comment. We have now removed this result of less relevance from the revised text.

3.15. "12 L116-126: The arguments here may not be well-based. Reference 16 does not relate to control of chronic conditions, but notes the very well known bias- the "healthy vaccinee effect". Those who are feeling unwell delay vaccination until they feel better. Also, guidelines suggest delaying vaccine after a positive SARS-CoV-2 test for a few days. The main morbidities here are hypertension and diabetes and it is not clear that control of these conditions affects receipt of a vaccine. Usual guidelines would suggest that those who are chronically unhealthy should receive vaccination but if that is not the case in Hong Kong then again the results may not be generalisable."

Author Response:

Thank you for pointing this out. The situation in Hong Kong is slightly different from many other parts of the world in that the older people and those living with chronic conditions show a higher degree of vaccine hesitancy. In fact, the vaccination rate among the age group of 80+ is as low as 16.9% as of 9 November 2021. Recommendations are those living with stable chronic conditions receive the vaccine and those with less well-controlled people consult a doctor. We have discussed our results may not be applicable to recipients have poor-controlled chronic conditions.

"The mass Covid-19 vaccination program in Hong Kong was launched on February 23, 2021 for CoronaVac and March 6, 2021 for Comirnaty. The first batch of residents eligible for vaccination consisted of healthcare workers, those aged 60 or above (and carers of those aged 70 or above), staff of care homes, essential public service providers, and cross-boundary transportation or port workers. The second batch (March 8) were food handlers, public transport service workers, construction workers, school teachers and staff, staff of the tourism industry. The third batch (March 16) further included people aged 30 – 59, students aged 16 or older studying abroad, as well as domestic helpers. Starting from April 15, other people

Response to Reviewer Comments

Manuscript Title: *Multimorbidity and adverse events of special interest associated with CoronaVac (Sinovac) and Comirnaty (Pfizer-BioNTech)*

Journal: *Nature Communications*

Ref. No.: *NCOMMS-21-35429*

aged 16 or above were made eligible. People were given the free choice of deciding whether or not to receive the vaccines, which one of the two vaccines to receive, as well as the venue of vaccination, but a different vaccine for the second dose was not allowed.²³ For Comirnaty, contraindications are hypersensitivity to previous doses of Comirnaty, or to the active substance or to any of the excipients. For CoronaVac, contraindications include a history of allergic reactions, severe underlying conditions, pregnancy, or lactation.”

(Lines 218-230, P. 7)

“This observation is in line with the official guidelines published by the Hong Kong Government only recommending that patients living with chronic conditions receive the vaccine if their conditions are under stable control.²⁶ Our results may not be applicable to recipients who had poorly controlled chronic conditions. In fact, the vaccination rate among the age group of ≥ 80 was as low as 16.9% as of November 9, 2021.²⁷”

(Lines 128-132, P. 5)

- 3.16. “13 L126: Might the phrase “If multimorbidity does imposes additional AESI risk increase following vaccination” be better as “If multimorbidity does impose additional AESI risk following vaccination”?”**

Author Response:

Thanks very much for this suggestion. We have replaced this phrase with your suggested expression.

“If multimorbidity does impose additional AESI risk following vaccination, the test for effect modification (interaction in Model 3) should still be able to detect this risk increase.”

(Lines 135-137, P. 5)

- 3.17. “14 L134-140: This is a very good point and might be emphasised more-not sure if it should be in the Abstract?”**

Author Response:

Thank you for your positive comment on this point of view and your suggestion. Please find it now inserted in the abstract as well.

“Prior research using electronic health records for Covid-19 vaccine safety monitoring typically focuses on specific disease groups and excludes

Response to Reviewer Comments

Manuscript Title: *Multimorbidity and adverse events of special interest associated with CoronaVac (Sinovac) and Comirnaty (Pfizer-BioNTech)*

Journal: *Nature Communications*

Ref. No.: *NCOMMS-21-35429*

individuals with multimorbidity, defined as ≥ 2 chronic conditions. We examined the potential additional risk of adverse events of special interest (AESI) 28 days after the first dose of vaccination with CoronaVac or Comirnaty imposed by multimorbidity.”

(Lines 30-33, P. 2)

- 3.18. “15 L147: as far as this reviewer can tell, reference 21 says nothing about vaccine hesitancy or concerns on safety being greater in those with multimorbidity. It is a paper about parental attitudes towards childhood vaccines and surely of limited relevance here?”**

Author Response:

Thank you for pointing this out. Please find it replaced with a more relevant study.

“Subject to further international research to replicate and verify our results, the implications of this study are important to reassure the public with regard to the widespread concern about vaccine safety among individuals living with multimorbidity who might be hesitant towards vaccine uptake.³²”

(Lines 160-162, P. 5)

- 3.19. “16 L153-155: As noted above, this argument seems mistaken- where the infection is under control vaccine coverage is already likely to be high, but the findings are surely relevant whatever the level of control of the infection. This point does not need to be made.”**

Author Response:

Thank you for this comment. We have now removed this point from the revised text.

- 3.20. “17 L155-156: the “third point” is unclear. The overall coverage seems to be 41% from Figure 1, but this probably varies a great deal with age. Why is it important to say that since most people do not receive the vaccine the absolute risk of AESIs is low? This seems to not be a sensible point.”**

Author Response:

Thank you for this comment. We have now removed this point from the Discussion.

Response to Reviewer Comments

Manuscript Title: *Multimorbidity and adverse events of special interest associated with CoronaVac (Sinovac) and Comirnaty (Pfizer-BioNTech)*

Journal: *Nature Communications*

Ref. No.: *NCOMMS-21-35429*

3.21. “18 L179: What is the nature of the ID used for matching?”

Author Response:

Thank you for this question. It was mainly the Hong Kong Identity Card Number but also could include passport numbers for foreign workers or visitors. Please find this detail clarified in the revised text.

“De-identified vaccination records provided by the Department of Health were linked by matching an encrypted unique person ID (Hong Kong Identity Card Number or foreign passport number) between the two databases. The Department of Health is the Government agency responsible for overseeing the implementation of mass vaccination in Hong Kong. Under the Prevention and Control of Disease (Use of Vaccines) Regulation (Cap 599K),²⁴ all Covid-19 vaccines administered are recorded with personal identification information in order to facilitate an efficient tracking of vaccine recipients in case of urgent safety issues.”

(Lines 203-209, P. 6)

3.22. “19 L188: How was information from 2005 to 2018 obtained? Is it that for those included between 2018 and 2021, their entire records going back to 2005 were obtained? When did the EHR system start? Has there been validation of the disease coding within the database over this period?”

Author Response:

Thank you very much. We have indeed been provided data on AESI history starting from 2005 for those active service users in 2018-2021 (used to remove those with AESI history for the detection of purely incident cases) and their chronic conditions records from 2018 – 2021 were used to identify the cohort. The computerized patient record system was launched in the early 1990s but earlier records were not made available to us for analysis. Previous works have examined the accuracy of the HA data and estimated a positive predictive value of over 85% for stroke and myocardial infarction. Please find these details now inserted in the revised text. We have now stated these details more clearly.

“We retrieved the records of patients who received inpatient or outpatient services provided by the HA between January 1, 2018 and July 31, 2021, and selected those ever coded with a diagnosis of any of 20 chronic conditions in their medical records since 2018 based on a widely used list of conditions for

Response to Reviewer Comments

Manuscript Title: *Multimorbidity and adverse events of special interest associated with CoronaVac (Sinovac) and Comirnaty (Pfizer-BioNTech)*

Journal: *Nature Communications*

Ref. No.: *NCOMMS-21-35429*

multimorbidity operationalization⁴¹ including hypertension, diabetes mellitus (type 2), severe constipation, depression, cancer, hypothyroidism, chronic pain, asthma, alcohol misuse, chronic pulmonary disease, schizophrenia, rheumatoid arthritis, peptic ulcer disease, cirrhosis, psoriasis, Parkinson's disease, dementia, irritable bowel syndrome, inflammatory bowel disease, and peripheral vascular disease using International Classification of Diseases, Ninth Revision (ICD-9) and International Classification of Primary Care, Second Edition (ICPC-2)."

(Lines 231-239, P. 7)

"We further removed those who died before the index date, were hospitalized on the index date, had their first chronic disease diagnoses only after the index date, or had AESI records before the index date. We retrospectively examined the medical records of the patients up to 2005 to avoid omitting any prior diagnosis of AESI."

(Lines 247-250, P. 7)

"These two sources of data have been used for previous Covid-19 vaccine safety research.^{7,12} Previous research validating the diagnostic codes of the HA data showed that for myocardial infarction and stroke, the positive predictive value was estimated above 85%³⁷. The HA databases have previously extensively used for drug safety and epidemiological studies.³⁸⁻⁴⁰ It is a longitudinal electronic medical record database allowing us to identify the newly recorded diagnoses."

(Lines 209-213, P. 6)

3.23. "20 L194: It is obviously difficult to deal with medical events and conditions that could contribute to the multimorbidity but may also be classed as AESI if they occurred after vaccination. It is not clear whether this is a particular problem with diabetes and coronary artery disease. Type I diabetes is stated to be an AESI and type II is a morbidity. Will it always be possible to be clear about this? Is the coding in the EHRs sufficiently good to be sure that events that are recorded after vaccination are truly newly incident events? Hypertension is strongly associated with CAD; can the authors be sure the CAD are incident events? Some more detail of which aspects of morbidity were excluded might be given unless it is simply the codes in eTable 8."

Author Response:

Response to Reviewer Comments

Manuscript Title: *Multimorbidity and adverse events of special interest associated with CoronaVac (Sinovac) and Comirnaty (Pfizer-BioNTech)*

Journal: *Nature Communications*

Ref. No.: *NCOMMS-21-35429*

Thank you for this comment. As we followed the list of AESI and the corresponding operational definitions, we did mainly use the ICD-9 codes to execute the exclusion. As mentioned above, the accuracy of the HA coding data is very good and it is a longitudinal electronic medical record database allowing us to identify the newly recorded diagnosis. Therefore, we are confident that the coding accuracy is adequate for a clear delineation of AESI and underlying chronic conditions. Please find the text revised with highlights on this accuracy. We have also acknowledged the reliance on the codes without further clinical investigation data to validate the diseases as a limitation.

“These two sources of data have been used for previous Covid-19 vaccine safety research.^{7,12} Previous research validating the diagnostic codes of the HA data showed that for myocardial infarction and stroke, the positive predictive value was estimated above 85%³⁷. The HA databases have previously extensively used for drug safety and epidemiological studies.³⁸⁻⁴⁰ It is a longitudinal electronic medical record database allowing us to identify the newly recorded diagnoses.”

(Lines 209-213, P. 6)

“Fourth, similar to other large scale pharmacovigilance studies using electronic medical record databases, we only relied on the diagnostic codes and other records for the operationalization of the diseases despite the demonstrated accuracy of those codes,³⁷ and specifically to distinguish between overlapping AESI and pre-existing chronic conditions.”

(Lines 197-190, P. 6)

- 3.24. “21 L205: Should the paper as a whole emphasise that it is just looking at first doses of vaccine (Censoring takes place at the second dose)? The main text of the paper does not make this clear- it continually refers to “vaccination”.”**

Author Response:

Thank you. We have now emphasized in the revised text that we focused on the first dose of vaccination.

“This study aims to examine the relationship between the first dose of Covid-19 vaccination and AESI among patients with chronic disease in Hong Kong and the potential additional AESI risk following vaccination associated with multimorbidity.”

(Lines 63-65, P. 3)

Response to Reviewer Comments

Manuscript Title: *Multimorbidity and adverse events of special interest associated with CoronaVac (Sinovac) and Comirnaty (Pfizer-BioNTech)*

Journal: *Nature Communications*

Ref. No.: *NCOMMS-21-35429*

“We found no evidence of a modified association between vaccination (first dose) and AESI among those living with multimorbidity compared with those without in general.”

(Lines 121-122, P. 4)

- 3.25. “22 L211; should it be “Receiving CoronaVac or receiving Comirnaty”? A reader might think patients received both vaccines when none did.”**

Author Response:

Thank you for the suggestion. Please find this part rephrased accordingly for better clarity.

“Exposure: receiving CoronaVac or receiving Comirnaty”

(Line 262, P. 8)

- 3.26. “23 L219-224: The use of entropy balancing is interesting, but it is unclear how with only binary covariates it differs from propensity score weighting? In addition, its usual use is with a binary treatment variable. Was the weighting method applied using vaccinated compared with unvaccinated? Did it involve a further step to compare the two vaccine groups? Was a conventional propensity score weighting used and if so what differences in results were obtained?”**

“24 L226-227: the entropy-balancing method usually relies on using logistic regression for estimation of causal effects. While it probably makes little difference, technically the Cox model has untestable assumptions when used to estimate causal effects. As noted above, there are several reasons for using an alternative analysis model. It is not clear what is meant by “vaccination status- it implies it is a comparison of vaccinated with unvaccinated, but it seems that model 1 (and the other models) use three treatment groups. This should be clearer.”

Author Response:

Thank you for this suggestion. We have now replicated the main analysis with the more commonly used propensity score-based weighting (with generalized boosted model) approach to check the robustness of the results. The results remained consistent without marked deviations. We also replaced ‘vaccination status’ with ‘vaccine groups’ to avoid confusion.

Response to Reviewer Comments

Manuscript Title: *Multimorbidity and adverse events of special interest associated with CoronaVac (Sinovac) and Comirnaty (Pfizer-BioNTech)*

Journal: *Nature Communications*

Ref. No.: *NCOMMS-21-35429*

“Eighth, we used propensity score-based generalized boosted model for an alternative weighting method for the analyses using the same R package, i.e. ‘WeightIt’.”

(Lines 299-301, P. 9)

3.27. “25 L243: There are very many possible packages able to be used for the analyses done here and to simply quote the “R statistical environment” is an inadequate description.”

Author Response:

Thank you very much. Please find the packages used for different purposes listed explicitly now.

“All analyses were conducted using the R statistical environment (Version 4.1.1, Vienna, Austria). For the weighting of the cohort, R package ‘WeightIt’ was used and ‘Survival’ was used for Cox regression.”

(Lines 303-304, P. 9)

3.28. “26 L244: Surely it is only assumed that there are no missing data? For each of the AESI for example, their presence is what is noted in the records, their absence is not actually noted but presumed.”

Author Response:

Thank you for this comment. Similar to all other pharmacovigilance study using electronic medical record database, there is an underlying assumption that only subjects with a record of diagnosis experience the adverse events. Subjects without a diagnosis are free from the adverse events; hence missing data category is not applicable. We have now noted that we only presumed no missing data with the absence of the relevant codes taken as the inexistence of such diagnoses.

“In common with all large-scale electronic medical record database research, we relied on the medical records for the operationalization of diseases and AESI, and it was presumed there were no missing data in our statistical analyses.”

(Lines 304-307, P. 9)

3.29. “27 Figure 1: the number of exclusions seems quite large and to have about 3% of the vaccinated patients getting a new chronic condition diagnosed in

Response to Reviewer Comments

Manuscript Title: *Multimorbidity and adverse events of special interest associated with CoronaVac (Sinovac) and Comirnaty (Pfizer-BioNTech)*

Journal: *Nature Communications*

Ref. No.: *NCOMMS-21-35429*

the 28 days after vaccination seems strange. The percentage seems a bit lower in the unvaccinated (possibly not unexpected) but still seems rather high. Is a comment necessary?"

Author Response:

Thank you very much. During our cohort selection, we inclusively identified individuals with a chronic condition diagnosis anytime since 2018 up to July 2021 for the age- and sex-matching and index date mapping (from the vaccinated to the unvaccinated) and then removed those who had their first chronic condition diagnosed only after the index date to exclude those who did not have 'pre-existing' chronic conditions.

The average age of our cohort is around 60 (SD=12), these are the age ranges within which people commonly receive diagnoses of various chronic diseases. Furthermore, we found that, before applying the age- and sex-matching and the exclusion criteria, 62.8% of the 'earliest' chronic condition diagnosis of each patient was identified in 2018, 17.8% in 2019, 11.3% in 2020, and 8.0% in 2021 (January to July). Accordingly, as the vaccination program was launched in February 2021, we think that approximately 3% of the patients being excluded due to a first diagnosis being after the index date is not unreasonably high. Please find this approach described more clearly in the revised text.

"We further removed those who died before the index date, were hospitalized on the index date, had their first chronic disease diagnoses only after the index date, or had AESI records before the index date. We retrospectively examined the medical records of the patients up to 2005 to avoid omitting any prior diagnosis of AESI."

(Lines 247-250, P. 7)

3.30. "28 Figure 2: it is usual for adverse events, to plot their cumulative frequency or cumulative hazard rather than a Kaplan-Meier survival curve. The figure for the Comirnaty treated patients seems strange. The truncation of the curve for those without multimorbidity at 22 days and the steep rise in the number of AESIs after 24 days for those with multimorbidity require some comment. The curves seem to be drawn smoothly but the data imply it should be a series of steps."

Author Response:

Response to Reviewer Comments

Manuscript Title: *Multimorbidity and adverse events of special interest associated with CoronaVac (Sinovac) and Comirnaty (Pfizer-BioNTech)*

Journal: *Nature Communications*

Ref. No.: *NCOMMS-21-35429*

Thank you very much for this suggestion. We have now replaced the Kaplan-Meier curves with cumulative incidence plots. The truncation at day 22 was mainly because of the censoring of the date of the second dose 21 days after the first (recommended dosing interval).

“Figure 2 shows the cumulative incidence of AESI by multimorbidity status and vaccine group over the follow-up period. Patients with multimorbidity were observed to have a faster increase of AESI incidence but no marked differences were identified between vaccine groups.”

(Lines 93-95, P. 4)

- 3.31. “29 Figure 3. The chord diagrams are an appealing way of displaying the data, but it could be good to have some extra eTables giving actual numbers.”**

Author Response:

Thank you very much, please find the eTable 2 added in the revised manuscript to show the underlying frequencies.

“The pairings were similarly patterned across all three groups, suggesting the co-occurrence of specific chronic conditions and AESI are similar between the unvaccinated, those receiving Comirnaty and those receiving CoronaVac. The underlying frequencies of the diagrams are shown in eTable 2.”

(Lines 97-100, P. 4)

- 3.32. “30 Table 1: I do not understand how the weighted numbers for the vaccinated groups are different from the unweighted? Surely the balancing is to make the unvaccinated similar to the vaccinated. Is it because the unweighted data is before matching which might have led to a lot of exclusions? Should there therefore be an extra column of the unweighted but matched data? This all seems rather puzzling.”**

Author Response:

Thank you very much for this comment. The entropy rebalancing technique does not use one of the matched groups as a referent to generate the weighting scores, but produces a series of appropriate weightings to achieve the highest possible balance between the groups. We have replicated the main analysis with the more commonly used propensity score-based weighting (with generalized

Response to Reviewer Comments

Manuscript Title: *Multimorbidity and adverse events of special interest associated with CoronaVac (Sinovac) and Comirnaty (Pfizer-BioNTech)*

Journal: *Nature Communications*

Ref. No.: *NCOMMS-21-35429*

boosted model) approach to check the robustness of the results. Similar findings were obtained. We have now also explained this difference in the revised text that report descriptive results.

“The entropy rebalancing technique we adopted produced a series of appropriate weights to achieve the highest possible balance between the three groups without using anyone of the groups as referent. After weighting, the maximum SMD for all covariates were all smaller than 0.1.”

(Lines 83-85, P. 3-4)

“Eighth, we used propensity score-based generalized boosted model for an alternative weighting method for the analyses using the same R package, i.e. ‘WeightIt’.”

(Lines 299-301, P. 9)

We hope these proposed revisions and additional analyses could sufficiently address all your concerns. We look forward to hearing from you again.

Sincerely,

Ian Chi Kei Wong, PhD

Corresponding author

Head and Professor

Department of Pharmacology and Pharmacy

Li Ka Shing Faculty of Medicine

The University of Hong Kong

REVIEWERS' COMMENTS

Reviewer #1 (Remarks to the Author):

Thank you for making these revisions, which have improved the paper. I still have concerns about the justification for this work, but other than that I think you have done a very good job in revising the paper.

Reviewer #2 (Remarks to the Author):

The authors have undertaken a thorough revision, including numerous re-analyses and sensitivity analyses. In my view they have addressed the reviewer comments and the paper makes a good contribution to the literature.

Reviewer #3 (Remarks to the Author):

NCOMMS-21-35429A

In general, the responses to the previous comments are reasonable and clear. However they clarify that there may be concerns with the generalisability of the findings and possibly also their reliability. In a number of places sensitivity analyses are reported as not changing conclusions but is there a reason why such analyses are not given in supplements (or it may be that I have failed to find them)?

It is now very clear that the vaccination roll-out policy in Hong Kong was possibly different to anywhere else in the world. Leaving out those who were most vulnerable seems to be a strange policy and is not clearly justified, but the authors are not responsible either for the policy or its justification. In some senses this is a paper whose purpose is to show that concerns that led to that perverse policy were not justified.

The point made that the investigation of adverse events effectively in the absence of Covid-19 disease does, in an "explanatory" sense allow for elucidation of adverse effects that may be related to the vaccines, means that this is a strength of the study is over-stated. It means that the generalisability of the findings to other settings may be very limited.

Points in the response that do not require further comment are omitted.

3.2 While, given the constraints of the Chinese governmental control of Hong Kong, this is a reasonable response, it cannot offer reassurance that e.g. the government has suppressed records of severe adverse reactions to its own vaccines. This may simply not be the case, but there has to be at least theoretical concern, but I am unsure how the authors can address this, so I sympathise with them.

3.7 The response has clarified things a great deal. It is clear that Hong Kong has a very different policy, not just a "slightly different" policy in regard to vulnerable people. This means that the results may or may not be generalisable. It is true to say that the randomised trials did not include many with multi-morbidities but most of those in public health did not regard that as a good reason to exclude them from vaccine rollout. Reference 14 notes this, and far from it showing that those with multi-morbidities are at higher risk from vaccines, it shows they are at higher risk from Covid (and other) disease, so the risk/benefit balance is likely to be even more favourable in them than in those without multi-morbidity. The issue of validity of this study is then raised because with such exclusions, the statistical power to detect effects may be low.

3.8 The papers cited are of very variable quality and clearly relate to adverse drug reactions.

Drug-drug interaction are clearly a concern in patients with multiple drugs and in those whose ability to metabolise drugs may be impaired by chronic kidney disease. It is reasonable to study adverse events in such people and to see if there is any evidence for particular problems with vaccines, but they are not drugs and their effects have very different mechanisms. The arguments for particular concern are weak, but it is certainly reasonable to check. The results found suggest that adverse events are consistently associated with multi-morbidity but neither with vaccines or a differential effect of vaccines in those with multi-morbidity. The authors are justified in looking for effects, but the emphasis on the expectation of such effects is not reasonable.

3.13 While this response is reasonable there are a number of places in the eTables 4 & 5 where there is complete nonsense output from the statistical models as a result of small numbers. This type of output should be suppressed otherwise it may be misinterpreted.

Response to Reviewer Comments

Manuscript Title: *Multimorbidity and adverse events of special interest associated with Covid-19 vaccines in Hong Kong*

Journal: *Nature Communications*

Ref. No.: *NCOMMS-21-35429A*

21 December 2021

Thank you very much for considering our manuscript favorably. We are thankful for another round of insightful reviews by the referees and your great effort in handling this submission. Please find appended below our detailed response to each of the specific remaining comments. The relevant text is quoted as appropriate:

Reviewer #1

- 1.1. “Thank you for making these revisions, which have improved the paper. I still have concerns about the justification for this work, but other than that I think you have done a very good job in revising the paper.”**

Author Response:

Thank you very much for your encouraging comment on the revisions. With regard to the justification of the study, it is well established that immune responses triggered by vaccination may induce inflammation to varying extents; hence it is possible such responses can increase the risk of serious adverse events because of the inflammation, thus necessitating cautious pharmacovigilance in these specific patient groups, who are at a higher risk of adverse health outcomes to begin with. We have now specified this general mechanism to motivate the study.

“Established evidence shows that varying degrees of inflammation may typically be induced by vaccination and the associated immune responses in general,²² cautious pharmacovigilance is indeed warranted for people with multimorbidity who are at a higher risk of adverse health outcomes to begin with.”

(Lines 54-57, P. 3)

Reviewer #2

- 2.1. “The authors have undertaken a thorough revision, including numerous re-analyses and sensitivity analyses. In my view they have addressed the**

Response to Reviewer Comments

Manuscript Title: *Multimorbidity and adverse events of special interest associated with Covid-19 vaccines in Hong Kong*

Journal: *Nature Communications*

Ref. No.: *NCOMMS-21-35429A*

reviewer comments and the paper makes a good contribution to the literature.”

Author Response:

Thank you very much for the positive overall comment on the revised manuscript.

Reviewer #3

3.1. “In general, the responses to the previous comments are reasonable and clear. However they clarify that there may be concerns with the generalisability of the findings and possibly also their reliability. In a number of places sensitivity analyses are reported as not changing conclusions but is there a reason why such analyses are not given in supplements (or it may be that I have failed to find them?)?”

Author Response:

Thank you very much for your overall positive comment on the revisions. Results of all the sensitivity analyses described in the Methods section have been provided in Supplementary Tables 6 – 14.

*“No marked deviations from the main analysis were observed in the findings of a series of sensitivity analysis although there were variations in the point estimates of the associations (**Supplementary Table 6 – Supplementary Table 14**). However, conclusions from the findings were not affected.”*

(Lines 117-119, P. 4)

We also acknowledge that with the vulnerable populations not being as actively vaccinated as other societies, there may be a limited generalizability of the results. Please find it now discussed as one of the limitations.

“Last, the fact that the vulnerable populations living with chronic conditions being vaccinated less proactively in Hong Kong compared with other societies may limit the generalizability of the findings. Also, as the population of Hong Kong is predominantly Chinese, replication of the analyses in other world populations is warranted.”

(Lines 189-192, P. 6)

Response to Reviewer Comments

Manuscript Title: *Multimorbidity and adverse events of special interest associated with Covid-19 vaccines in Hong Kong*

Journal: *Nature Communications*

Ref. No.: *NCOMMS-21-35429A*

- 3.2. “It is now very clear that the vaccination roll-out policy in Hong Kong was possibly different to anywhere else in the world. Leaving out those who were most vulnerable seems to be a strange policy and is not clearly justified, but the authors are not responsible either for the policy or its justification. In some senses this is a paper whose purpose is to show that concerns that led to that perverse policy were not justified.”**

Author Response:

Thank you for recognizing this additional implication of our study – given the absence of safety signals in the use of Covid-19 vaccines among this vulnerable population living with chronic conditions, the vaccination of these people should be prioritized. Please find it now inserted in the Discussion.

“Given the fact that people with multimorbidity have a higher risk of developing life-threatening complications if infected with SARS-CoV-2,³⁴ our results should be reassuring and support the notion that multimorbidity does not impose additional risk of AESI following vaccination and that the vaccination of these vulnerable citizens should be prioritized.”

(Lines 162-166, P. 5)

- 3.3. “The point made that the investigation of adverse events effectively in the absence of Covid-19 disease does, in an “explanatory” sense allow for elucidation of adverse effects that may be related to the vaccines, means that this is a strength of the study is over-stated. It means that the generalisability of the findings to other settings may be very limited.”**

Author Response:

Thank you for this comment. It has now been removed as a strength of the study.

- 3.4. “3.2 While, given the constraints of the Chinese governmental control of Hong Kong, this is a reasonable response, it cannot offer reassurance that e.g. the government has suppressed records of severe adverse reactions to its own vaccines. This may simply not be the case, but there has to be at least theoretical concern, but I am unsure how the authors can address this, so I sympathise with them.”**

Author Response:

Response to Reviewer Comments

Manuscript Title: *Multimorbidity and adverse events of special interest associated with Covid-19 vaccines in Hong Kong*

Journal: *Nature Communications*

Ref. No.: NCOMMS-21-35429A

Thank you very much for this comment. We understand the reviewer's concern. In our previous study, we were able to identify the increased risk of Bell's palsy associated with CoronaVac (see reference no. 7 in manuscript) which had been widely reported in the press and media without any interference from the Government. Indeed, the Hong Kong Government has demonstrated an open attitude towards the potential risk of adverse events related to the vaccines. For instance, it launched a compensation scheme for severe adverse reactions related to the vaccines (https://www.covidvaccine.gov.hk/en/AEFI_Fund) and disseminates the cumulative numbers of adverse events following immunization on a regular basis for the sake of transparency (<https://www.covidvaccine.gov.hk/en/dashboard/safety/aefi>). We hope our team's experience can reassure the reviewer in this regard. As this information does not naturally flow in the content of the manuscript, we have decided not to amend the manuscript.

- 3.5. **“3.7 The response has clarified things a great deal. It is clear that Hong Kong has a very different policy, not just a “slightly different” policy in regard to vulnerable people. This means that the results may or may not be generalisable. It is true to say that the randomised trials did not include many with multi-morbidities but most of those in public health did not regard that as a good reason to exclude them from vaccine rollout. Reference 14 notes this, and far from it showing that those with multi-morbidities are at higher risk from vaccines, it shows they are at higher risk from Covid (and other) disease, so the risk/benefit balance is likely to be even more favourable in them than in those without multi-morbidity. The issue of validity of this study is then raised because with such exclusions, the statistical power to detect effects may be low.”**

Author Response:

Thank you for this comment. A conclusion of reference 14 was that “*it is ... essential to monitor the effectiveness, safety, and immunogenicity of COVID-19 vaccines in the most vulnerable categories*”. We therefore cited it to highlight that safety is among the important concerns with regard to the vaccination of vulnerable populations.

We agree with the reviewer that, given the higher risk of serious complications following Covid-19, the risk/benefit balance is likely to be even more favorable for the vaccination of this vulnerable population.

Response to Reviewer Comments

Manuscript Title: *Multimorbidity and adverse events of special interest associated with Covid-19 vaccines in Hong Kong*

Journal: *Nature Communications*

Ref. No.: *NCOMMS-21-35429A*

“Of note, people living with multimorbidity are at increased risk of serious complications following an infection of SARS-CoV-2.21 It is currently unclear if multimorbidity is related to a risk increase of any AESI following Covid-19 vaccination.”

(Lines 57-59, P. 3)

We also acknowledge that with the vulnerable populations not being as proactively vaccinated in Hong Kong as in other societies, there may be a limited generalizability of the results. Please find it now discussed as one of the limitations.

“Last, the fact that the vulnerable populations living with chronic conditions being vaccinated less proactively in Hong Kong compared with other societies may limit the generalizability of the findings. Also, as the population of Hong Kong is predominantly Chinese, replication of the analyses in other world populations is warranted.”

(Lines 189-192, P. 6)

- 3.6. “3.8 The papers cited are of very variable quality and clearly relate to adverse drug reactions. Drug-drug interaction are clearly a concern in patients with multiple drugs and in those whose ability to metabolise drugs may be impaired by chronic kidney disease. It is reasonable to study adverse events in such people and to see if there is any evidence for particular problems with vaccines, but they are not drugs and their effects have very different mechanisms. The arguments for particular concern are weak, but it is certainly reasonable to check. The results found suggest that adverse events are consistently associated with multi-morbidity but neither with vaccines or a differential effect of vaccines in those with multi-morbidity. The authors are justified in looking for effects, but the emphasis on the expectation of such effects is not reasonable.”**

Author Response:

Thank you for this comment. We agree that drug-drug interactions may not constitute enough justification for this study. Nevertheless, it is well established that immune responses triggered by vaccination may induce inflammation to varying extents; hence it is possible such responses can increase the risk of serious adverse events because of the inflammation, thus necessitating cautious pharmacovigilance in these specific patient groups who

Response to Reviewer Comments

Manuscript Title: *Multimorbidity and adverse events of special interest associated with Covid-19 vaccines in Hong Kong*
Journal: *Nature Communications*
Ref. No.: *NCOMMS-21-35429A*

are at a higher risk of adverse health outcomes to begin with. We have now specified this general mechanism to motivate the study.

“Established evidence shows that varying degrees of inflammation may typically be induced by vaccination and the associated immune responses in general,²² cautious pharmacovigilance is indeed warranted for people with multimorbidity who are at a higher risk of adverse health outcomes to begin with.”

(Lines 54-57, P. 3)

- 3.7. “3.13 While this response is reasonable there are a number of places in the eTables 4 & 5 where there is complete nonsense output from the statistical models as a result of small numbers. This type of output should be suppressed otherwise it may be misinterpreted.”**

Author Response:

Thank you very much for this comment. We have now suppressed results for AESI outcomes with only five or fewer observed cases in any of the vaccination groups.

“We did not estimate the HR for AESIs of which only five or fewer cases occurred in any of the vaccine groups to avoid misinterpretation.”

(Lines 114-115, P. 4)

We hope these proposed revisions and additional analyses could sufficiently address the remaining concerns raised by the reviewers.

We look forward to hearing from you again.

Sincerely,

Ian Chi Kei Wong, PhD
Corresponding author

Head and Professor
Department of Pharmacology and Pharmacy
Li Ka Shing Faculty of Medicine
The University of Hong Kong